



# Flexural and compressive strength of the landfast sea ice in the Prydz Bay, East Antarctic

Qingkai Wang[1], Zhaoquan Li[1], Peng Lu[1], Yigang Xu[2], Zhijun Li[1],

[1] State Key Laboratory of Coastal and Offshore Engineering, Dalian University of Technology, Dalian, 116024, China

[2] Marine Design and Research Institute of China, Shanghai, 200011, China

*Correspondence to*: Zhijun Li (lizhijun@dlut.edu.cn)

**Abstract.** A total of 25 flexural and 55 uniaxial compressive strength tests were conducted using landfast sea ice samples collected in the Prydz Bay. Three-point bending tests were performed at ice temperatures of –12 to –3 ℃ with force applied vertically to original ice surface, and compressive tests were performed at –3 ℃ with a strain-rate level of $10^{-6}$–$10^{-2}$ s$^{-1}$ in the

directions vertical and horizontal to ice surface. Judging from crystal structure, the ice samples were divided into congelation ice, snow ice, and a mixture of the these two. The results of congelation ice showed that the flexural strength had a decreasing trend depending on porosity rather than brine volume, based on which a mathematical equation was established to estimate flexural strength. Both flexural strength and effective modulus increased with increasing platelet spacing. The uniaxial compressive strength increased and decreased with strain rate below and above the critical regime, respectively, which is 8.0

$\times 10^{-4}$–$1.5 \times 10^{-3}$ s$^{-1}$ for vertically loaded samples and $2.0 \times 10^{-3}$–$3.0 \times 10^{-3}$ s$^{-1}$ for horizontally loaded samples. A drop off in compressive strength was shown with increasing sea ice porosity. Consequently, a model was developed to depict the combined effects of porosity and strain rate on compressive strength in both ductile and brittle regimes. The mechanical strength of mixed ice was lower than congelation ice, and that of snow ice was much weaker. To provide a safe guide for the transportation of goods on landfast sea ice in the Prydz Bay, the bearing capacity of the ice cover is estimated with the lower

and upper envelopes of flexural strength and effective modulus, respectively, which turned out to be a function of sea ice porosity.

## 1 Introduction

Sea ice flexural strength, effective modulus, and uniaxial compressive strength are important ice engineering properties. They are always used to assess the ice load exerted on marine structures in ice-infested waters (Sinsabvarodom et al., 2020; Su et

al., 2010) and the load that can be supported by the floating ice (Masterson, 2009). Scientific and commercial activities have been expanding in the polar regions in recent years (Arctic Council, 2009; Mayewski et al., 2005). Therefore, studies on sea ice mechanical properties are still required.

Sea ice flexural strength and effective modulus are obtained simultaneously using two different bending tests: cantilever and simple beam tests (Ji et al., 2011; Karulina et al., 2019). The cantilever beam test is a full-scale measurement performed on a



large ice beam with whole thickness through the ice cover. The simple beam test (three- or four-point supported) is carried out using cuboid samples cut free from the ice cover. Because of maintaining intact ice state, the cantilever beam test gives in situ ice flexural strength. However, it is highly time- and labor-consuming to prepare sufficient full-thickness cantilever beams, especially in the polar regions. Compared with bending test, the uniaxial compression test is relatively easier to be carried out because of the smaller size of samples required. Moreover, the behavior of ice under compression tests is affected by machine

stiffness (Sinha and Frederking, 1979). So, the test is performed using a machine with high-stiffness loading frame (Bonath et al., 2019; Moslet, 2007).

The mechanical strength of sea ice has a strong dependence on its physical properties, to be more exact, on the multiphase structure. Timco and O'Brien (1994) compiled a database of 939 reported measurements on the flexural strength of sea ice from polar and temperate regions, and proposed a widely used empirical equation of sea ice flexural strength relying on brine

volume fraction. Recently, a new formula of flexural strength dependent on brine volume was reported in Karulina et al. (2019), by performing a series of full-thickness cantilever beam tests. The parameterization of sea ice effective modulus based on brine volume fraction was also given in Karulina et al. (2019). While investigations showed that the gas within sea ice may occupy more space than brine, especially for warm ice (Frantz et al., 2019; Wang et al., 2020), and an overestimation will be produced by calculating strength only with brine volume. Sea ice strength should depend more accurately on the total porosity (Timco

and Weeks, 2010). The studies of sea ice flexural strength and effective modulus related to porosity are rare. On the contrary, previous researches have related sea ice uniaxial compressive strength to porosity (Kovacs, 1997; Moslet, 2007; Timco and Frederking, 1990). The commonly adopted formulae to estimate sea ice uniaxial compressive strength were proposed by Timco and Frederking (1990), where 283 small-scale strength tests performed mostly in the Arctic waters were collected. A limitation in their equations is that the applicable condition is only for strain rate less than $10^{-3}$ s$^{-1}$ corresponding to ductile strain-rate

regime in their report.

Compared with Arctic sea ice, the understanding of mechanical properties of Antarctic sea ice is limited because of booming oil and gas exploration in the Northern Hemisphere polar regions in the last century. With current expansion in science and tourism in the south pole, sound knowledge of Antarctic sea ice engineering properties is urgent. It is thought that the ice strength depends on the conditions of ice cover formation and its development, which in turn determines the ice structure.

Unlike Arctic sea ice, there is a large fraction of granular ice layer in the Antarctic sea ice (Carnat et al., 2013; Jeffries et al., 2001). Therefore, the empirical equations established based on Arctic sea ice strength tests may not be appropriate for Antarctic sea ice. Furthermore, because of the heterogeneous variability shown by sea ice in different Antarctic seas responding to climate change (Hobbs et al., 2016; Matear et al., 2015), sea ice in different sea areas may also behave differently in mechanical properties.

The landfast sea ice area is the only corridor for scientific expeditions to transport logistics cargos to the research stations in the shore of Prydz Bay. However, the available observations on the landfast sea ice conditions in the Prydz Bay is quite limited (Hui et al, 2017; Zhao et al., 2020), let alone, the sea ice mechanical properties that are of importance to ensure safe activities on ice. The full-scale ice trial of an icebreaker was conducted in the landfast sea ice in the Prydz Bay in the late Austral spring



in 2019. A full-thickness ice block was extracted through the ice cover, part of which was used for on-site measurements, and
the rest was stored for subsequent detailed investigations on sea ice mechanical properties in domestic laboratory. In this paper,
we present the results of various mechanical experiments performed in laboratory, including crystal structure, flexural strength,
effective modulus, and uniaxial compressive strength of the Antarctic landfast first-year sea ice. The results will help to deepen
the understanding of mechanical properties of Antarctic sea ice, especially the landfast sea ice in the Prydz Bay.

## 2 In situ sampling and laboratory experiments

### 2.1 In situ sampling

As part of the 36th Chinese National Antarctic Research Expedition, a site (69.2 ºS, 76.3 ºE) was selected on the landfast sea
ice (Fig. 1a) in the Prydz Bay to carry out ice sampling for ice trial of the vessel in November 2019. A large ice block with
sectional dimensions of 1.0 m × 1.2 m was extracted through the whole ice cover with 1.6 m in thickness covered by 0.2 m
thick snow (Fig.1b). Salinity measurements of an ice core beside the ice block gave a "C-shaped" depth profile with a bulk
value of 5.2±1.3 psu, indicating that it was a first-year ice. After lifting onto the deck using the ship crane, part of the ice block
was cut and machined into seven three-point beams from the bottom to top, giving a flexural strength of 718.6±47.6 kPa
measured using the same machine as adopted in the laboratory tests (Fig. 1c, see details in Section 2.2.2). The rest of the ice
block was packed in plastic bags to avoid sublimation and stored at a temperature of –20°C to reduce brine loss. Throughout
the five-month storage, they were shipped to Dalian, China for detailed mechanical experiments in laboratory.

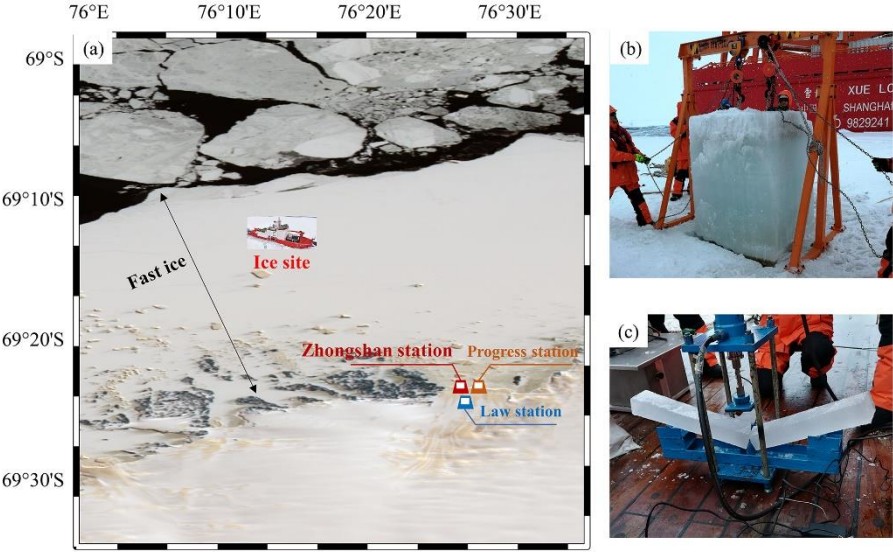


**Figure 1: (a) Maps of the ice sampling site (69.2 ºS, 76.3 ºE), (b) ice block extracted for measurements, (c) bending tests performed
on ship deck. The background in (a) is the ice image on 23 November 2019 derived from the HY-1C satellite
(https://osdds.nsoas.org.cn/).**



## 2.2 Laboratory experiments

### 2.2.1 Crystal structure

Vertical and horizontal sections were made for ice crystal structure measurements. One cm thick sections at approximately 10 cm intervals along the ice thickness were prepared first using a band saw. These thick sections were attached to glass plates and thinned down further to an approximate 2 mm thickness. A planer was then used to reduce the thickness to 0.5 mm. The thin slices were placed on a universal stage to observe the crystal structures under crossed polarized light, recorded by

photography. Image sizes were calibrated using a ruler with resolution of 1 mm. Moreover, the images of horizontal slices were further analyzed for grain sizes using an image processing software. The ice grains were distinguished separately and then were recognized as circles to determine the diameters from areas, all of which in a horizontal section were finally averaged.

### 2.2.2 Bending test

The flexural strength and effective modulus of sea ice were measured using three-point supported beam tests. The rough-cut

ice beams were prepared using chain saw, which were then machined carefully to section dimensions of 7 cm ×7 cm using band saw. The long axis of the beam was in the horizontal plane of the original ice cover. Cuboid-shaped ice samples were finally obtained using the band saw at right-angles to the beam sides to make the length 65 cm. Afterwards, they were persevered in a thermotank at required temperatures for at least 24 hours before experiments.

The three-point bending test was conducted in the cold laboratory using equipment as shown in Fig. 2. The device was powered

by a hydraulic actuator. A stainless-steel column was fixed on the bottom of pressing plate to give a line force on the midspan of the ice beam underlain by a simply supported frame with a span of 60 cm. A force sensor with a capacity of 500 N and an accuracy of ±0.25 N was used to record the load. A laser sensor (accuracy ±12.5 μm) attached to the device columns and an aluminum plate fixed onto the pressing plate to reflect the laser from the sensor were used together to record the displacement of the plate, i.e. the deflection at the middle of the ice beam. Both force and displacement were recorded at frequencies of 200

Hz. The time-of-loading was typically below 10 s with few extending to less than 30 s.

Based on the linear elasticity theory, the flexural strength of ice ($\sigma_\mathrm{f}$) is defined as in Eq. (1).

$$\sigma_\mathrm{f} = \frac{3Fl}{2bh^2} \tag{1}$$

where $F$ is force at failure, $l$ is span between supports, $b$ and $h$ are section width and height of the ice beam, respectively. The effective modulus of ice ($E$) is Eq. (2).

$$E = \frac{Fl^3}{4bh^3\delta} \tag{2}$$

where $\delta$ is displacement of the pressing plate at failure.



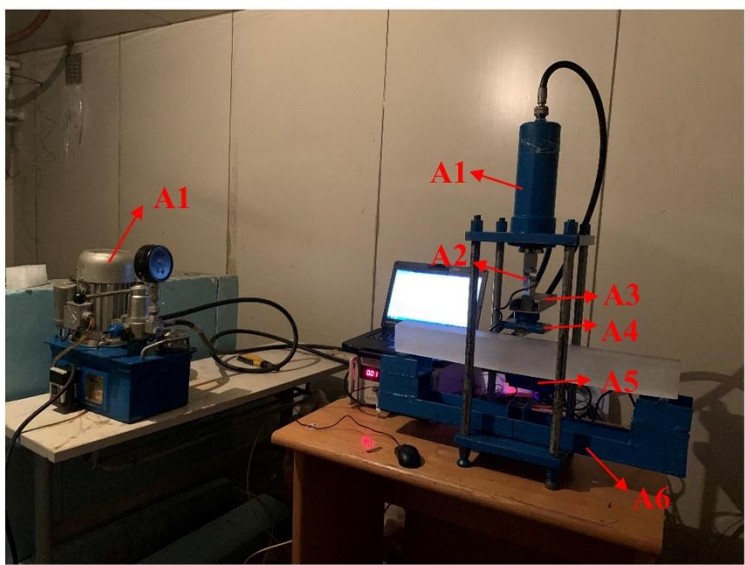

**Figure 2: Experiment equipment for the three-point bending test: (A1) hydraulic system; (A2) force sensor; (A3) aluminum plate; (A4) line-load pressing plate; (A5) laser displacement sensor (behind the ice sample); (A6) simply supported frame.**

Before loading, the mass of each sample was weighed using a balance (±0.1 g) and the volume was calculated based on the dimensions measured using a caliper (±0.02 mm). After failure, half of the broken sample was collected to melt for salinity measurements using a salinometer (±0.001 psu), and the remaining was used to observe ice grain size and ice platelet spacing. The ice platelet spacing was measured using the same method as grain size. The brine volume fraction and porosity of the ice beam were calculated using the ice temperature, salinity, and density based on Cox and Weeks (1983). A total of 25 bending

tests were performed, and Table 1 shows the conditions of bending tests. Details of the ice samples can be seen in Table A1 in Appendix A.

**Table 1: The test conditions of bending and compression experiments.**

| Test | Temperature (℃) | Strain rate (s$^{-1}$) | Loading direction |
|---|---|---|---|
| Bending test | –12, –8, –5, –3 | / | Vertical to original ice surface |
| Compression test | –3 | $10^{-6}$–$10^{-2}$ | Parallel and vertical to original ice surface |

### 2.2.3 Compression test

Similar to sample preparation of bending tests, cuboid-shaped compression samples with dimensions of 7 cm × 7 cm × 17.5 cm were finally prepared with long directions horizontal and vertical to original ice surface, respectively. Both ends of the samples were planed using a spoke shave with care taken to keep them flat and made perpendicular with long axis by checking with a square rule. After preparation, the samples were also stored in a thermotank at required temperatures for at least 24 hours before compression.



Because of the higher-rigidity set up required by the compression test as compared to the bending test, a universal testing machine with a servo motor was used, which can maintain a constant loading rate with an accuracy of ±0.5% (Fig. 3). The machine is equipped with a force sensor of 100 kN capacity and ±0.5% accuracy as well as a displacement sensor with ±2 μm accuracy. Both force and displacement were recorded at frequencies of 50 Hz. The ice sample was compressed in a cryostat in which cold source was provided by a refrigerating system. The mass and dimensions of each sample were measured before

compression and the fragments were collected to melt for salinity measurements after failure. The porosity of each ice sample was also determined according to Cox and Weeks (1983). Crystal texture observations were not performed as the samples generally broke into small pieces or with large deformation. A total of 55 compression tests were performed, including 28 vertically loaded samples and 27 horizontally loaded samples. The test conditions are concluded in Table 1, and details are listed in Tables A2 and A3 in Appendix A. The uniaxial compressive strength of ice ($\sigma_c$) is given by Eq. (3).

$$\sigma_c = \frac{F}{S} \tag{3}$$

where $S$ is section area of the compression sample. The strain rate of the compression sample ($\dot{\varepsilon}$) is defined as in Eq. (4).

$$\dot{\varepsilon} = \frac{\dot{\delta}}{L} \tag{4}$$

where $L$ is the length of compression sample, and $\dot{\delta}$ is the loading rate of pressing plate.

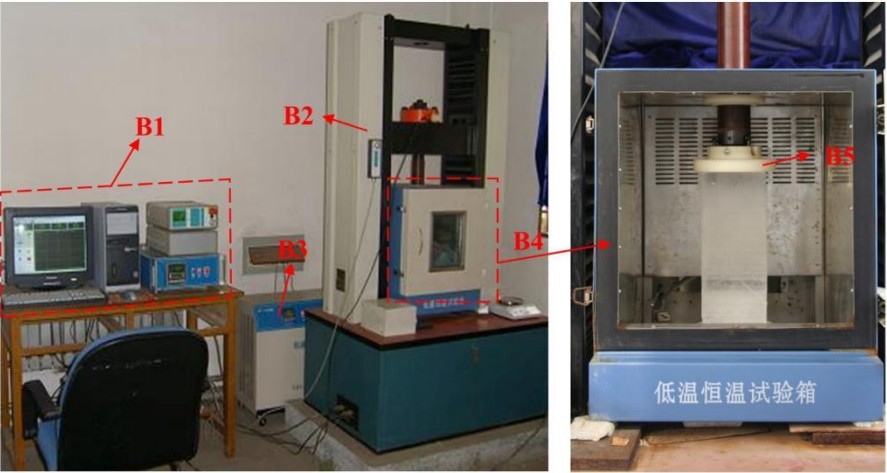

**Figure 3: Experiment equipment for the uniaxial compression test: (B1) control system; (B2) loading frame; (B3) refrigerating system; (B4) cryostat container; (B5) pressing plate.**

**2.2.4 Uncertainty analysis**

Uncertainty due to measurement error can be estimated with an error propagation analysis. For flexural strength, it is given by Eqs. (5) and (6).



$$\frac{\Delta \sigma_{\mathrm{f}}}{\sigma_{\mathrm{f}}} = \left| \frac{\partial \ln \sigma_{\mathrm{f}}}{\partial F} \right| \Delta F + \left| \frac{\partial \ln \sigma_{\mathrm{f}}}{\partial b} \right| \Delta b + \left| \frac{\partial \ln \sigma_{\mathrm{f}}}{\partial h} \right| \Delta h \tag{5}$$

$$\frac{\Delta \sigma_{\mathrm{f}}}{\sigma_{\mathrm{f}}} = \frac{\Delta F}{F} + \frac{\Delta b}{b} + \frac{2\Delta h}{h} \tag{6}$$

where $\Delta(\cdot)$ is the errors of corresponding parameters.

Therefore, the total uncertainty of the flexural strength is approximately 0.2%. Using the same method, the uncertainty of the effective modulus estimated by error propagation is approximately 2.9%, and that of compressive strength is approximate 0.6%
and of strain rate is approximately 0.5%. The inherent scatter in the test results is considerably greater than the uncertainty caused by system.

## 3 Results

### 3.1 Crystal structure

Results of ice crystal measurements are shown in Fig. 4. There was a snow-ice layer in the top 28 cm of the ice sheet where a
large number of fine-grained crystals were in the shape of small polygons, shown in the vertical and horizontal sections. The snow ice could also be judged by its white appearance. Their diameters were too small (< 1 mm approximately) to be identified precisely. Below the top snow-ice layer there was an ice layer with grains elongated perpendicularly with respect to ice surface, which was formed by congelation of sea water without disturbance. Judging from the cross sections of congelation ice layer, it can be divided further into transition- and columnar-ice layers. The transition ice existed just below the snow-ice layer at a
depth of 28–40 cm, in which grains were fine-grained with small diameters ≈ 2 mm. The exact diameters were also difficult to obtain due to excessive quantities and small sizes. Below the transition layer, there was columnar ice with larger grain size of 0.7–1.2 cm.



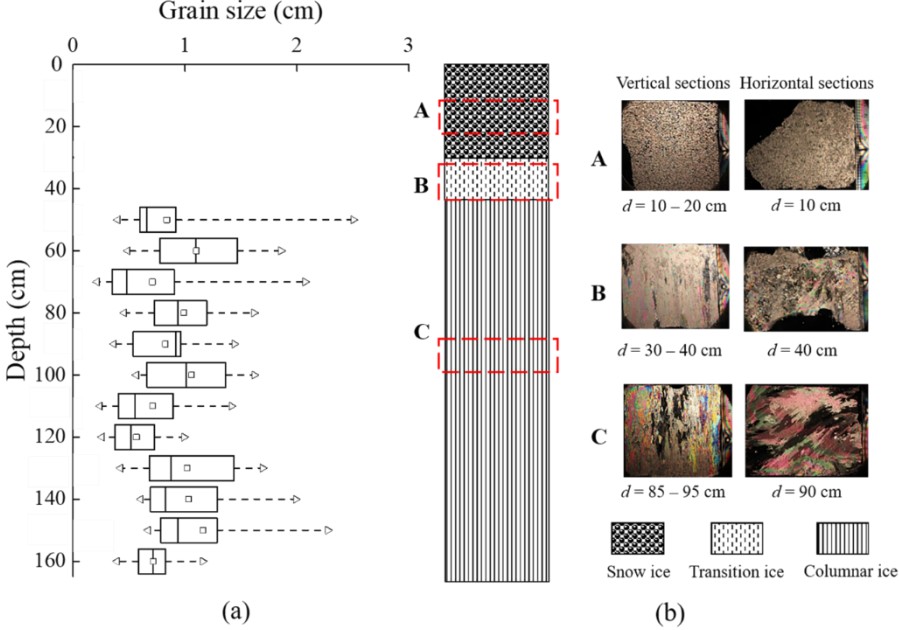

**Figure 4: (a) Box chart of grain sizes along ice depth. The grain sizes in the top 40 cm of ice sheet were too small to be identified clearly. (b) The stratigraphy diagram of ice crystal structure profile with typical pictures of grain types where _d_ represents ice depth.**

### 3.2 Flexural strength

A total of three types of ice samples were measured in the bending tests which consisted of snow ice, columnar ice, and a mixture of transition and columnar ice. The snow-ice specimen could be distinguished easily by their white appearance and light weight. The mixed-ice specimen showed distinct stratification, of which the transition-ice part looked milky in appearance with 2 cm thickness. Figure 5 shows the typical curves of applied force varying with deflection at the middle of the simple beam for different types of ice samples. The force increased linearly with deflection and dropped abruptly when ice broke. Columnar- and mixed-ice samples behaved similarly, while for snow-ice beam, the force and deflection at failure were much less than the others. The failure modes of the ice beams are also given in Fig. 5, where the cracks (black lines) were particularly prominent, and the snow-ice sample also showed a main crack penetrating through the samples at the middle as shown by the others. A total of 16 columnar-ice beams and five mixed-ice beams were conducted, respectively. In terms of strength, the flexural strength of mixed ice ranged from 511.3 to 845.9 kPa with an average of 687.9±153.2 kPa, similar to the strength of columnar ice ranging from 305.3 to 1119.7 kPa with an average of 698.8±222.6 kPa. Therefore, the flexural strength of columnar and mixed ice was analyzed together in Section 3.2.1 termed congelation ice and that of snow ice was analyzed separately in Section 3.2.2.



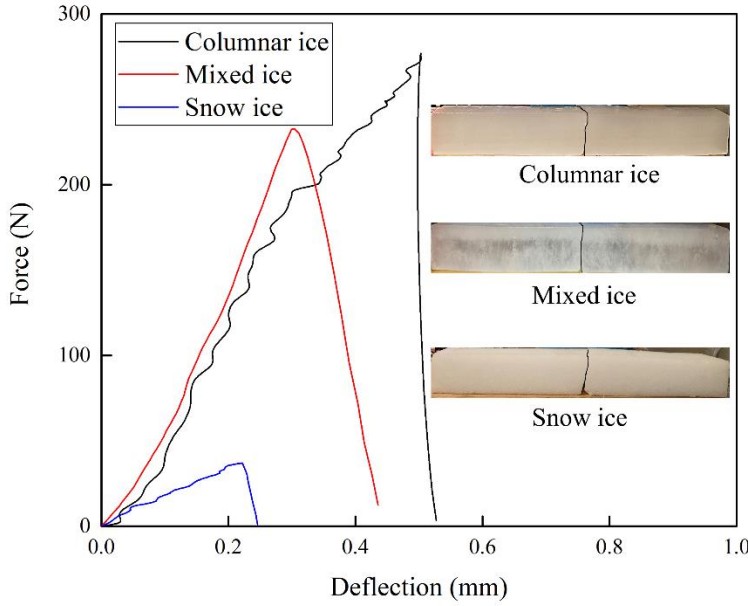

**Figure 5: Typical curves of force versus deflection at the middle of beam with corresponding broken sections depicted by black lines shown in subplot.**

### 3.2.1 Congelation ice

To validate the dependence of flexural strength of our samples on sea ice brine volume, the brine volume fractions of ice samples were determined and taken to the previously established equations listed in Table 2. Results shown in the subplot in Fig. 6 indicate that the equation of Karulina et al. (2019) underestimated our flexural strength by 30% approximately on average. The difference may contribute to the region-specific ice conditions. The experiments in Karulina et al. (2019) was performed near Svalbard archipelago in the Arctic Ocean, and different ice conditions of ice cover formation and development determined different ice strength. Additionally, their sea ice flexural strength was derived from cantilever beams, and stress concentrations at the root of beam as well as more flaws contained in larger-size beams result in low strength. The equation provided by Timco and O'Brien (1994) was derived from various regions and test approaches, and thus its estimation agreed better with our data than Karulina et al. (2019). Even so, it was still conservative and overestimated by approximately 20% than that of ours. To further quantify the dependence of flexural strength on brine volume fraction of our ice samples, regression analysis was conducted not only using the mathematical expressions as given in Table 2, but also other commonly used functions. However, all the fitting equations were accompanied with low determination coefficients ($R^2$) less than 0.1 and were not significant even at 0.1 significance level ($p$).





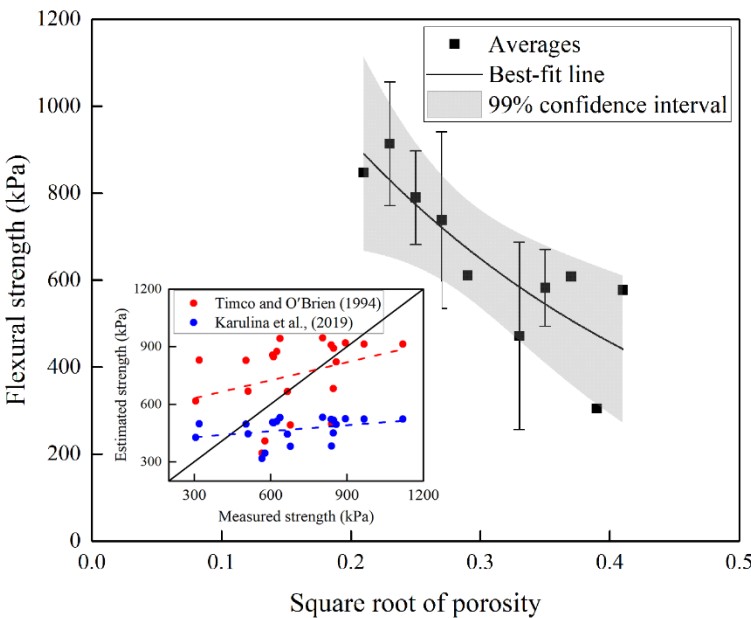

**Figure 6: The relationship between average sea ice flexural strength and porosity for congelation ice. The subplot shows the comparisons between estimations using previously established equations and measured strength.**


**Table 2: Empirical relationships of flexural strength and effective modulus based on brine volume fraction ($v_b$). $\sigma_f$ is in MPa and $E$ is in GPa.**

| Reference | Equation |
| --- | --- |
| Timco and O'Brien, 1994 | $\sigma_f = 1.76\exp(-5.88\sqrt{v_b}) \ \left(\sqrt{v_b} \leq 0.5\right)$ |
| Karulina et al., 2019 | $\sigma_f = 0.5266\exp(-2.804\sqrt{v_b})$ |
| | $E = 3.1031\exp(-3.385\sqrt{v_b}) \ \left(\sqrt{v_b} \leq 0.5\right)$ |

Alternatively, flexural strength was related to sea ice porosity. Taking the square root of porosity of 0.02 as a bin, the mean

value and standard deviation were determined, and the relationship between flexural strength and porosity is shown in Fig. 6. Adopting the same forms as that for brine volume fraction, a best-fit relationship was obtained using regression analysis with $R^2 = 0.68$ and $p < 0.01$. The equation was depicted as Eq. (7).

$$\sigma_f = 1859.06\exp(-3.51\sqrt{v}) \ \left(0.2 < \sqrt{v} < 0.5\right)$$

(7)

where $\sigma_f$ is in kPa and $v$ is porosity.





The bending failure of ice is considered as a tensile failure in brittle manner, and grain size affects ice failure under tension. Consequently, to investigate the relationship between flexural strength and grain size, the grain size of columnar ice was divided at an interval of 0.1 cm, and the mean strength was determined. A clear trend is shown in Fig. 7a in which flexural strength of columnar ice decreased with increasing grain size. For a homogeneous crack-free polycrystalline ice with the same scale of grain size as our samples, the fracture process at a high strain rate is nucleation-controlled. The crack propagates as

soon as it forms, and the stress at failure is inversely proportional to grain size (Sanderson, 1988). It was noteworthy that probably because of the minor effect of grain size compared with porosity, the best-fit trend was not significant at $p = 0.1$. Therefore, it is difficult to draw a firm conclusion on the dependence of flexural strength on grain size. Schulson (2001) argued that the case for sea ice may be more complicated, as brine pockets can act as pre-cracks, and platelet spacing might be more important than grain size. Therefore, the mean strength of columnar ice was determined at a platelet spacing interval of 0.01

cm, and the relationship between flexural strength and platelet spacing was plotted in Fig. 7b. Results showed that the flexural strength increased with increasing platelet spacing, and the regression analysis gave best-fit trend with a higher $R^2$ at $p = 0.05$. It is thus comprehensible that the larger the platelet spacing, the less are the existing pre-cracks for propagation.

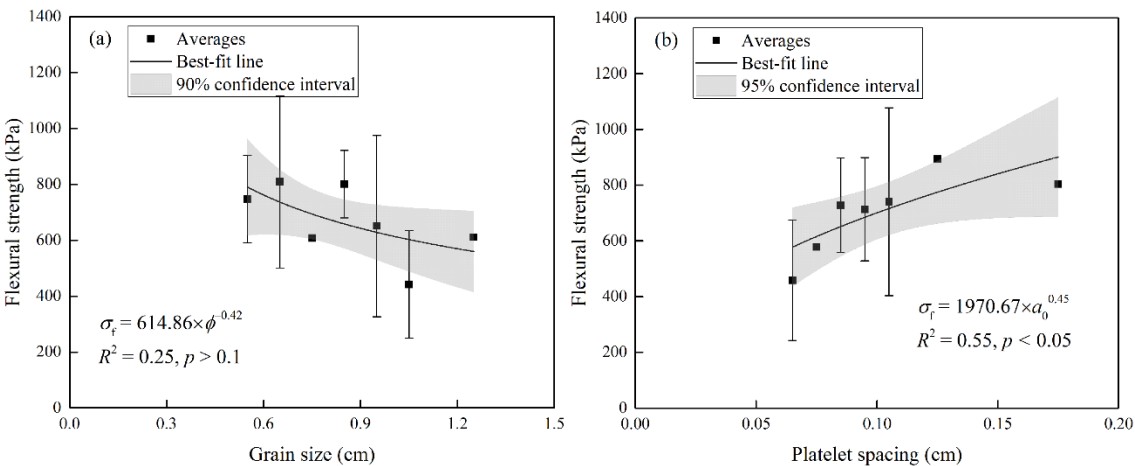

**Figure 7:** The variations of average flexural strength with respect to (a) grain size and (b) platelet spacing for columnar ice. Also
shown are the best-fit equations, where $\phi$ and $a_0$ are grain size and platelet spacing, respectively.

### 3.2.2 Snow ice

A total of four snow-ice beams were tested, and results showed that the flexural strength of snow ice was much weaker than that of congelation ice, with an average of 122.7±37.3 kPa. One of the snow-ice beams was tested at a temperature of –5 ℃ and showed relatively stronger flexural strength (176.9 kPa) than the mean value of the others (104.6±11.1 kPa) tested at a

temperature of –3 ℃.





### 3.3 Effective modulus

#### 3.3.1 Congelation ice

The effective modulus of columnar-ice samples ranged from 0.4 to 2.3 GPa, with an average of 1.5±0.5 GPa, and that of mixed ice ranged from 0.9 to 2.0 GPa with an average of 1.6±0.5 GPa. Because similarities were shown between the effective modulus
of columnar and mixed ice, they were also analyzed together termed congelation ice.

The effective modulus of our ice samples has no statistically significant dependence on brine volume fraction. Karulina et al. (2019) have proposed a mathematical relationship between sea ice effective modulus and brine volume fraction, in which the effective modulus decreased with the square root of brine volume fraction in an exponential manner (Table 2). Compared with our measurements, the calculation using their formula gave 1.5 times overestimation. Considering the effect of sea ice porosity
on the flexural strength, the dependence of effective modulus on porosity was investigated. However, regression analysis indicated a weak relationship between them with low $R^2$ and no statistical significance at $p = 0.1$ level using the form proposed by Karulina et al. (2019) and other commonly used ones.

The effects of sea ice sub-structure on the effective modulus of columnar ice were investigated as shown in Fig. 8, where the relationships between effective modulus and grain size as well as platelet spacing were examined using the same processing
approaches as flexural strength. Results showed that there was a negligible effect of grain size on effective modulus with $R^2$ close to 0 (Fig. 8a). On the contrary, the effect of platelet spacing on effective modulus was significant. The effective modulus increased with increasing platelet spacing (Fig. 8b), and regression analysis showed that the logarithmic equation was the best-fit form to depict the varying trend.

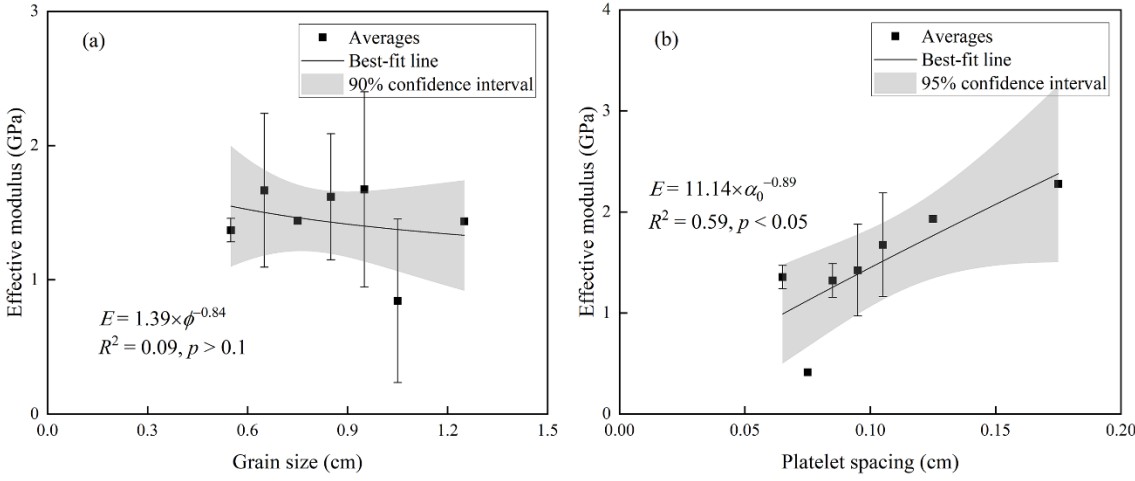

**Figure 8: The variations of average effective modulus with respect to (a) grain size and (b) platelet spacing for columnar ice. Also shown are the best-fit equations.**





### 3.3.2 Snow ice

The mean effective modulus of snow ice was 0.4±0.1 GPa, much lower than that of congelation ice. Ice temperature may have a slight effect on the snow-ice effective modulus. Of the snow-ice samples, the one tested at a temperature of –5 ℃ had an
effective modulus of 0.4 GPa, which is similar with the average modulus (0.4±0.2 GPa) of the others tested at a temperature of –3 ℃.

### 3.4 Uniaxial compressive strength

The uniaxial compressive samples were divided into three types, i.e. congelation-ice samples consisting of columnar or/and transition ice, snow-ice samples, and mixed-ice samples consisting of both snow and congelation ice. Of the three types, the
compressive strength of congelation ice is much stronger than the other two, so it was analyzed separately.

### 3.4.1 Congelation ice

Figure 9 shows the typical stress-strain curves of sea ice during compression tests. Different behaviors were shown by sea ice compressed under different strain rates and loading directions. At a low strain rate, stress increased linearly with strain until peak, and then decreased gently without abrupt change, indicating ductile behavior (curves B and D). Large deformation with
local cracks can be seen in the samples at failure (samples B and D). With an increase in the strain rate, sea ice exhibited brittle behavior, where the stress dropped abruptly once reaching peak (curves A and C). No obvious deformation occurred at the time of failure (samples A and C). In spite of similar stress-strain curves, there were several differences between vertically loaded and horizontally loaded samples at failure. For the vertically loaded samples compressed under a low strain rate (sample B), cracks developed along the long axes of the columnar grains; while for horizontally loaded samples (sample D), inclined
cracks developed at both ends of the ice. For horizontally loaded samples, force was applied perpendicular to the long axes of the ice columns, and the sliding along the grain boundary was easily triggered, combined with the dislocation pile-up at grain boundaries, leading to crack development along the grain boundaries. While for vertically loaded samples, force acted on the cross of the columns, and grain boundary sliding was suppressed. Therefore, local cracks developed parallel to the column axes due to the grain decohesion. Alternatively, at a high strain rate, local cracks and deformation were not sufficient to relax
the stress concentration. Therefore, for the vertically loaded samples, once the sea ice was split by a vertical crack, the slender columns suddenly became unstable and failed through buckling (sample A). For horizontally loaded samples, inclined cracks accumulated and formed a primary crack penetrating through the sample, and the sample failed through shear faulting (sample C).



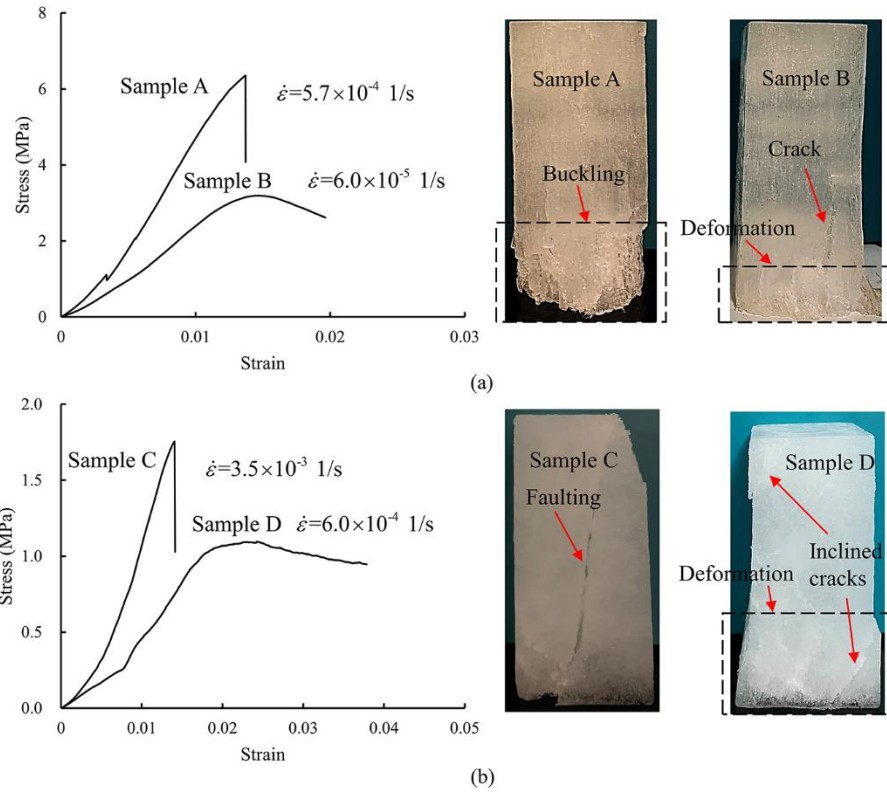

**Figure 9: Typical stress-strain curves and failure modes of (a) vertically and (b) horizontally loaded samples.**

Since the test strain-rate was wide ranging from $10^{-6}$ to $10^{-2}$ s$^{-1}$, the average strength was not able to be determined in most bins if the strain rate range was divided into intervals. Therefore, all the measured data was put in Fig. 10a with double logarithmic coordinates, showing the relationship between uniaxial compressive strength and strain rate. At the strain rates where ice breaks in a ductile manner (termed ductile strain-rate regime), the uniaxial compressive strength increased with increasing strain rate. The power law is commonly adopted to describe the variations of uniaxial compressive strength with strain rate at the ductile strain-rate regime (Timco and Weeks, 2010). The best-fit equations for vertically and horizontally loaded strength were given, and both showed higher $R^2$ at $p = 0.01$. At a brittle strain-rate regime, the uniaxial compressive strength exhibited strain-rate weakening; while regarding the mathematic equation to depict the declining trend of strength at brittle regime, no agreement has yet been reached. Here, the power-law equation was still adopted, and regression analysis gave statistically significant results with high $R^2$. When sea ice transits from ductile to brittle failure, it reaches maximum uniaxial compressive strength. In addition, because sliding along the grain boundary is easily triggered when force is applied perpendicular to the long axes of the columns, the uniaxial compressive strength of vertically loaded samples was higher than that of horizontally loaded samples at both brittle and ductile regimes.





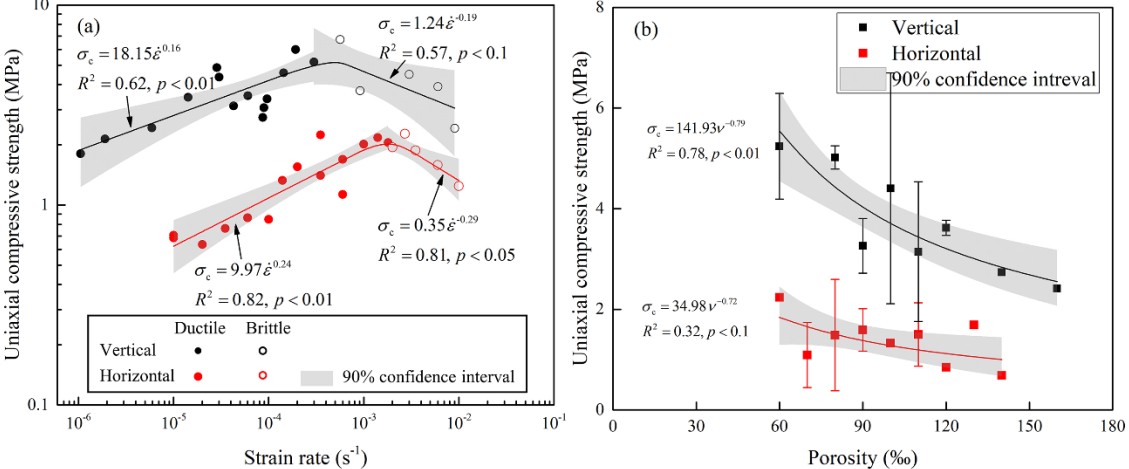

**Figure 10: Variations of uniaxial compressive strength with strain rate (a) and porosity (b) for congelation ice. Also shown the best-fit equations. Figure (a) adopts all measured data and Fig. (b) adopts averages determined taking the porosity of 10‰ as a bin. Also show are the best-fit lines and equations.**

The mean strength and standard deviation were determined taking the porosity of 10‰ as a bin, and the variations of uniaxial compressive strength with respect to porosity are plotted in Fig. 10b. As the sea ice porosity increases, strength decreases since brine and gas inclusions cannot support the load. Regression analysis showed a statistically significant dependence of uniaxial compressive strength on porosity following a power law.

Since the uniaxial compressive strength of sea ice is affected by strain rate and porosity, it is rational to parameterize the compressive strength using these two parameters. Previous researchers have already related the uniaxial compressive strength of sea ice to porosity and strain rate. The frequently used parameterization was proposed by Timco and Frederking (1990), which has been validated by the Arctic sea ice strength data; while it is uncertain whether their model is appropriate for the Antarctic sea ice. Overestimates were obtained by taking our test data into the previously established model, and the ratio was $1.42 \pm 0.25$ for vertically loaded samples and $1.78 \pm 0.45$ for horizontally loaded samples. Furthermore, the applicable strain-rate range was below $10^{-3}$ s$^{-1}$ in Timco and Frederking's model, corresponding to a ductile regime in their report, and no parameterization was given for the strength at the brittle regime. Another model of sea ice uniaxial compressive strength was also proposed in Kovacs (1997), which related the horizontally loaded sea ice strength in the rate-strengthening regime to porosity using the form of Eq. (8). Overestimates of $1.27 \pm 0.36$ were obtained using the Kovacs's model, and the regional specific conditions of sea ice may contribute to the difference.

$$\sigma_c = A\dot{\varepsilon}^B v^C$$

(8)

where $A$, $B$, and $C$ are fitting coefficients.





Based on the aforementioned relationships between strain rate, porosity, and uniaxial compressive strength, the form of Eq. (8) was adopted here to estimate the sea ice uniaxial compressive strength, and extended to cover both ductile and brittle regimes simply by changing the coefficients (Table 3).

**Table 3: The fitting coefficients of Eq. (8).**

| Loading direction | Ductile strain-rate regime | | | | Brittle strain-rate regime | | | |
| --- | --- | --- | --- | --- | --- | --- | --- | --- |
| | $A$ | $B$ | $C$ | $R^2$ | $A$ | $B$ | $C$ | $R^2$ |
| Vertical | 325.75 | 0.13 | −0.70 | $0.77^\alpha$ | 7.88 | −0.26 | −0.48 | 0.76 |
| Horizontal | 153.87 | 0.25 | −0.58 | $0.93^\alpha$ | 2.97 | −0.34 | −0.52 | $0.85^\beta$ |

α and β represent significance level of 0.01 and 0.1, respectively.

All the regressions showed good $R^2$ at $p = 0.1$ or higher level, except that of vertically loaded strength at brittle regime. Here, the form of Eq. (8) was applied because of the respective effects of strain rate and porosity on the uniaxial compressive strength. Statistically significant agreements between estimations and test data for horizontally loaded uniaxial compressive strength prove the applicability of this model. Probably due to the limited data points, the fit to vertically loaded strength at brittle regime was not significant, which requires more measurements for validation in future study.

The three-dimensional surfaces of sea ice uniaxial compressive strength varying with strain rate and porosity are plotted in Fig. 11. It was found that sea ice transits from ductile to brittle behavior over a range of strain rates. For vertically loaded samples, the ductile-to-brittle range is approximately $8.0 \times 10^{-4}$–$1.5 \times 10^{-3}$ s$^{-1}$, and that for horizontally loaded samples is narrower by $2.0 \times 10^{-3}$–$3.0 \times 10^{-3}$ s$^{-1}$. These transition strain rates agreed well with the typical ranges given by Schulson (2001). Indeed, the transition is a result of competition between stress relaxation and stress build-up (Schulson, 2001). Under vertical loading, the stress concentration is relaxed by grain decohesion, and for horizontal loading, it is relaxed by grain boundary sliding. Because the instantaneous elastic response of the decohesion is more sensitive to the strain, ductile-to-brittle strain rate under horizontal loading is higher (Ji et al., 2020). Moreover, the transition strain rate increased with increasing porosity. Brine and gas inclusions in sea ice can act us pre-cracks promoting stress concentration. On the other hand, with sea ice being more porous, both grain decohesion and grain boundary sliding is much easily triggered. It seems that the effect of porosity on the grain boundary sliding is relatively slight, so that the transition range is narrow for horizontal loading.



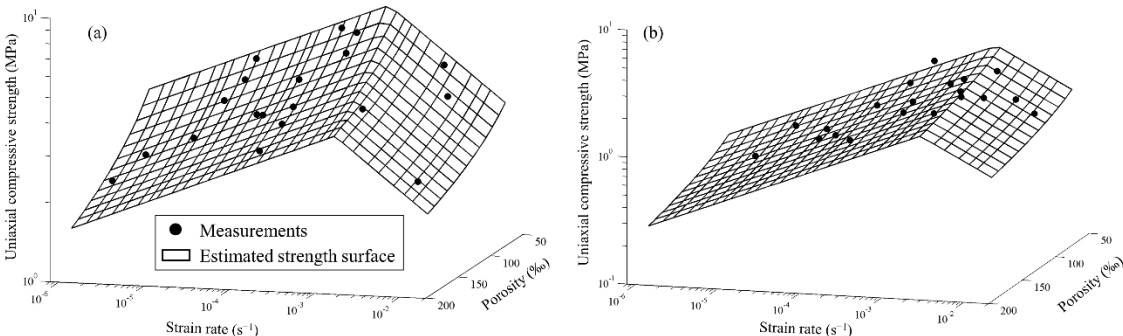

**Figure 11: Three-dimensional surfaces of uniaxial compressive strength of (a) vertically and (b) horizontally loaded samples varying with strain rate and porosity.**

### 3.4.2 Mixed and Snow ice

A total of four mixed-ice samples were compressed under vertical loading direction, of which two were tested at $2.0 \times 10^{-4}$ s$^{-1}$ and the others were tested at $2.0 \times 10^{-3}$ s$^{-1}$. But several phenomena were still found from the limited tests. The stress-strain curves and failure modes of ice samples compressed under different strain rates were different. At the low strain rate of $2.0 \times 10^{-4}$ s$^{-1}$, which is lower than the ductile-to-brittle strain rate of vertically loaded congelation ice, stress showed an approximate yield stage after reaching peak, during which the part of snow ice had a large deformation (Fig. 12a). While at the high strain rate of $2.0 \times 10^{-3}$ s$^{-1}$, which is greater than the transition strain rate of vertically loaded congelation ice, stress drops suddenly after peak and the whole ice sample failed through buckling of ice slender columns (Fig. 12b). The mean strength of mixed ice samples compressed at the low strain rate was only a bit higher than that at the high strain rate ($1.7\pm0.2$ vs. $1.1\pm0.1$ MPa); while both were weaker than that of congelation ice at corresponding strain rate.





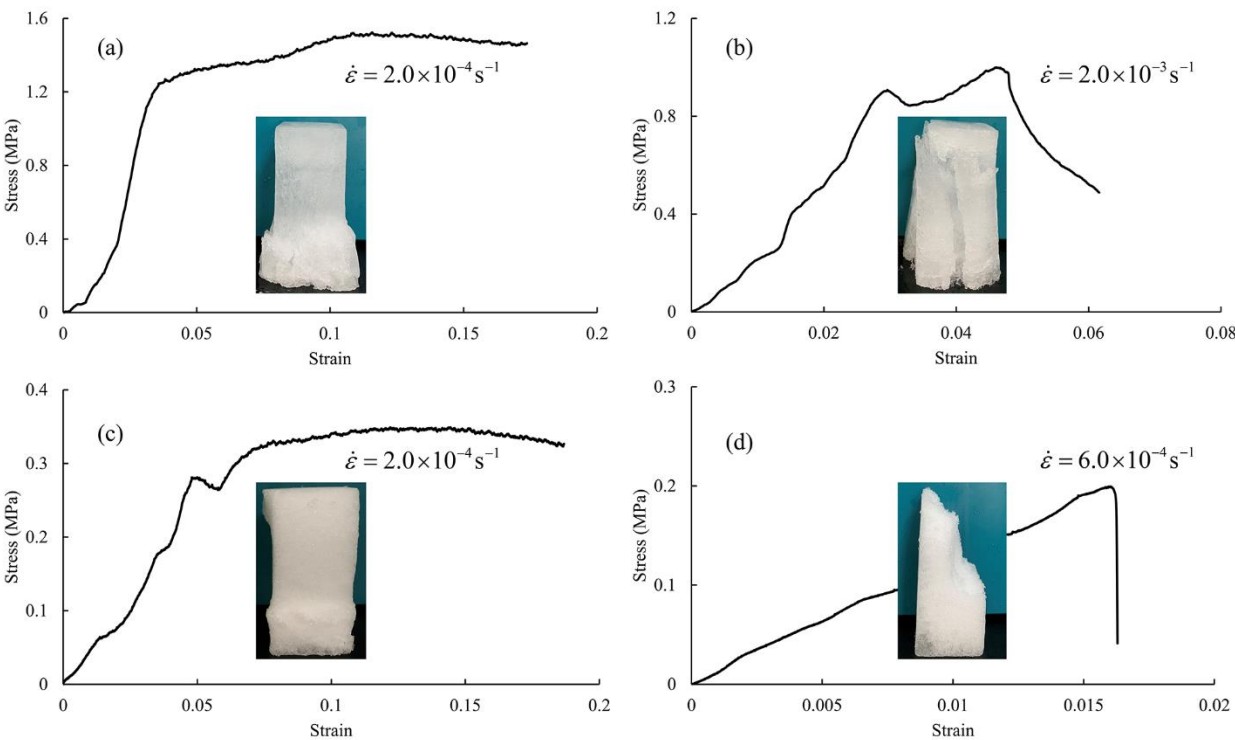

**Figure 12: Typical stress-strain curves and failure modes of mixed ice samples (a) and (b) as well as of snow ice samples (c) and (d) under vertical loading direction.**

Similarly, snow ice samples compressed at different strain rates also showed distinct stress-strain curves. Taking the vertically

loaded snow ice samples as examples, at low strain rates (e.g. $2.0 \times 10^{-4}$ $s^{-1}$), creep deformation occurred when the curve

exhibited a yield stage (Fig. 12c). The samples seemed to be compacted judging from its appearance, but the bulk density only

increased by 1.03. At high strain rates (e.g. $6.0 \times 10^{-4}$ $s^{-1}$), snow ice broken into parts and lost its strength suddenly (Fig. 12d).

Both vertically and horizontally loaded snow ice samples had the same behaviors and similar strength values at low and high

strain rates. The mean strength was 0.3±0.1 MPa for both types of samples with creep deformation, which was higher than that

for samples with sudden failure with values of 0.1±0.1 MPa.

## 4 Discussion

### 4.1 The ratios between sea ice strength

From the view of field operation, it is easier to obtain the vertically loaded uniaxial compressive strength than the horizontally

loaded compressive strength and flexural strength, because ice cores can be removed from ice cover directly using an ice driller.

However, the latter two properties of sea ice are more useful than the former in engineering applications. Since both

compressive strength and flexural strength are affected by porosity, taking the porosity of 10‰ as a bin, ratios were obtained





by comparing the mean strength located in the same porosity interval. It was found that the ratio of vertically loaded uniaxial compressive strength averaged at 3.1±0.9, and that of vertically loaded uniaxial compressive strength to flexural strength at 7.4±1.9. Consequently, it is possible to estimate the horizontally loaded uniaxial compressive strength and flexural strength just by performing vertically loaded compressive tests if on-site conditions are not allowable for conducting tests. Similar comparisons have also been reported in Timco and Frederking (1990) that the uniaxial compressive strength of vertically loaded columnar ice is generally 1–4 times higher than horizontally loaded columnar ice, and the ratio increased with decreasing porosity.

**4.2 Comparison between field measured and empirical estimated flexural strength**

In Section 3.2.1, a mathematical equation Eq. (7) was proposed to determine the flexural strength of landfast sea ice in the Prydz Bay based on sea ice porosity. In addition to field measurement of sea ice salinity (see Section 2.1), sea ice temperature was also measured immediately after extracting the ice core nearby the ice block. A negative-gradient layer was found in the top 10 cm with ice temperatures from –2.3 to –3.5 ℃ from the top down, and underneath was a positive-gradient layer with an ice temperature of –1.9 ℃ in the bottom. The mean ice temperature was –2.6±0.6 ℃. Since no ice density measurements were conducted in the field, it is assumed as the typical value of first-year ice of 0.92 g cm$^{-3}$ according to Timco and Weeks (2010). As a result, the bulk porosity of the ice block extracted is determined to be 107‰, and the sea ice flexural strength is then estimated as 591.3 kPa with an interval of 470.9–713.7 kPa at $p = 0.01$. The field measured strength is located in the estimated confidence interval, but almost reaches the upper limit.

There are several uncertainties in comparisons between flexural strength measured on site and empirically estimated. First, the temperature gradient in ice when testing onboard was probably different with its in situ state after removal and machining. Since the air temperature was similar with the bulk ice temperature, the onboard measured sea ice flexural strength was possibly stronger than its in situ strength. Second, the brine drainage during ice core sampling produced underestimation of ice salinity measurements (Notz et al., 2005), and consequently leading to an overestimated sea ice porosity and underestimated flexural strength evaluation. But, the underestimation of ice salinity by a core-based measurement is unknown, making it difficult to quantify the effect on the estimated flexural strength. The brine drainage also occurred at the bottom during ice block extraction. While this kind of brine loss does not correspond to phase equilibrium, and will not change the fraction of pure ice; therefore, sea ice strength is not affected. Change of sea ice strength is caused by ice temperature change after lifting, which influences the sea ice phase composition.

Overall, the empirical model Eq. (7) depicts the general trend of sea ice flexural strength varying with porosity. One of the methods to improve the estimation accuracy is to reduce the brine drainage during ice salinity measurement. Notz et al. (2005) developed a nondestructive instrument of ice salinity based on electrical impedance measurements, but it is not suited for pre-existing ice and must be deployed in sea water. On the other hand, the cantilever beam test can keep in situ ice temperature gradient in ice. Therefore, combining more reliable ice salinity measurements with the cantilever beam test results is a potential way to produce a more effective empirical equation in future studies.





## 4.3 Bearing capacity of landfast sea ice in the Prydz Bay

Due to the severe ice conditions and unknown offshore water depth, icebreakers usually have to stop in the landfast sea ice area in the Prydz Bay (Fig. 1), and the logistics cargos for research stations have to be transported by helicopters and trucks shortly after they are unloaded on ice. Therefore, it is of paramount importance to estimate the ice bearing capacity to ensure safety. A typical scenario of unloading cargos from an icebreaker on landfast sea ice in the Prydz Bay is shown in Fig. 13. The channel broken by the icebreaker acts as a wet crack penetrating the ice, i.e. the entire ice side surface is exposed to water (Masterson, 2009), and the cargos are loaded not far from the ship. The bending of the ice under the load causes flexural stress to be imposed on the ice cross section. If the maximum flexural stress does not exceed the ice strength, the load will be supported. Following the industry standard required by ISO19906 (2019), the extreme fiber stress in a cracked ice sheet due to a uniformly distributed load is predicted by Eqs. (9) and (10).

$$\sigma_{\mathrm{f}} = 0.529(1+0.54\upsilon)\frac{P}{H^2}[\lg(\frac{EH^3}{kr^4})-0.71] \tag{9}$$

$$r = \begin{cases} \sqrt{1.6c^2+H^2}-0.675H & \text{for } c < 1.742H \\ c & \text{for } c \geq 1.742H \end{cases} \tag{10}$$

where $E$ and $\sigma_{\mathrm{f}}$ are in kPa; $\upsilon$ is Poisson ratio $\approx 0.33$; $P$ is magnitude of load in kN; $H$ is ice thickness; $k$ is foundation modulus $= 9.81$ kPa m$^{-1}$; $r$ is effective beam length; and $c$ is the radius of loaded area.

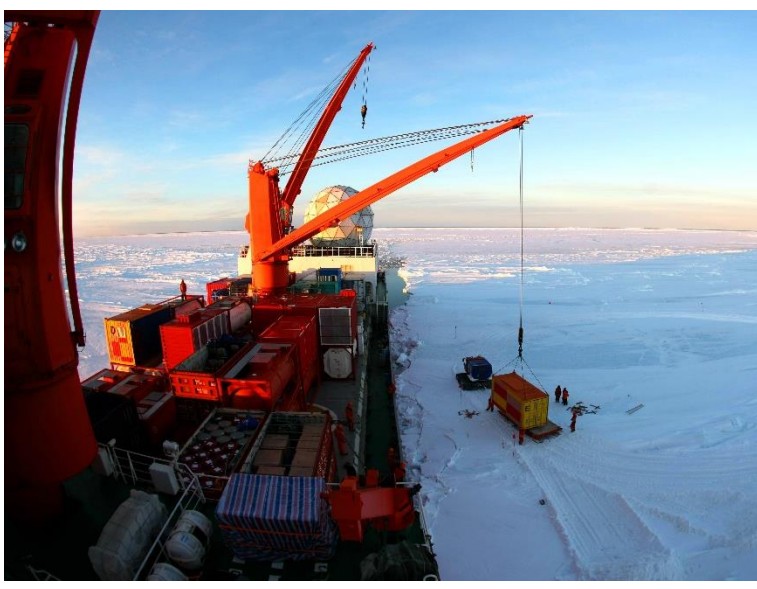

**Figure 13: Scenario of unloading cargos from an icebreaker on landfast sea ice in the Prydz Bay (Photted by Shiping Liu).**

Noticeably, the flexural strength calculated using Eqs. (9) and (10) is based on the elastics beam theory assumption. Kerr and Palmer (1972) deduced the distributions of bending stresses considering varied elastic modulus along ice thickness in a floating





ice plate. It is difficult to obtain the real distribution of elastic modulus along ice thickness, and Eqs. (9) and (10) have been proved reliable by experience (ISO19906, 2019), and hence, are adopted here. For the purpose of safe designing, a more

conservative prediction is preferred in practice. According to Eq. (9), the bearing capacity increases with increasing flexural strength and decreasing effective modulus. Therefore, minimum flexural strength and maximum effective modulus are required. Section 3 talks about the variations of flexural strength and effective modulus with porosity in terms of averages. So, all the measured data were plotted in Fig. 14a and b to determine the lower and upper envelopes of flexural strength and effective modulus, respectively, providing the best-fit equations given by Eqs. (11) and (12).

$$\sigma_{\mathrm{f,min}} = 54.90\exp(-1.76\sqrt{v}) \ \ (R^2 = 0.97, \ p < 0.01) \tag{11}$$

$$E_{\max} = 7.23\exp(-4.20\sqrt{v}) \ \left(R^2 = 0.77, \ p < 0.05\right) \tag{12}$$

where $\sigma_{\mathrm{f,min}}$ is the minimum flexural strength, and $E_{\max}$ is the maximum effective modulus.

As stated in Section 3.1, the flexural strength of snow ice is much weaker than congelation ice, making the snow cover negligible in terms of bearing capacity. Therefore, the ice thickness was taken as 1.3 m, which is the thickness of congelation-

435 ice layer of the ice block based on the crystal texture (Fig. 4). Additionally, Eqs. (9) and (10) worked reliably when the loaded

radius was not large enough compared with the characteristic length ($L_c$) of sea ice ( $L_c = \left[ \dfrac{EH^3}{12k(1-v^2)} \right]^{\frac{1}{4}}$ ). With sea ice porosity

increasing from 40 to 260‰, the characteristic length decreased from 16.0 to 11.6 m. Therefore, several different loaded radiuses were selected within the characteristic-length range, and the magnitude of load was expressed as a function of sea ice porosity in Fig. 14c by substituting Eqs. (11) and (12) into Eqs. (9) and (10). The unit of load is transformed to tons in Fig.

14c necessitated by practitioners (e.g. sea captain) for better understanding. With loaded radius increasing from 2 to 10 m, the level of load increased from 2 to 10 t approximately. With loaded radiuses of 2, 4, 6, and 8 m, the bearing capacity of ice decreased by 32.4%, 26.9%, 19.6%, and 7.9%, respectively, with increasing porosity from 40 to 260‰. On the contrary, with a large loaded radius of 10 m, the bearing capacity of ice increased by 15.4% with porosity increasing from 40 to 260‰.





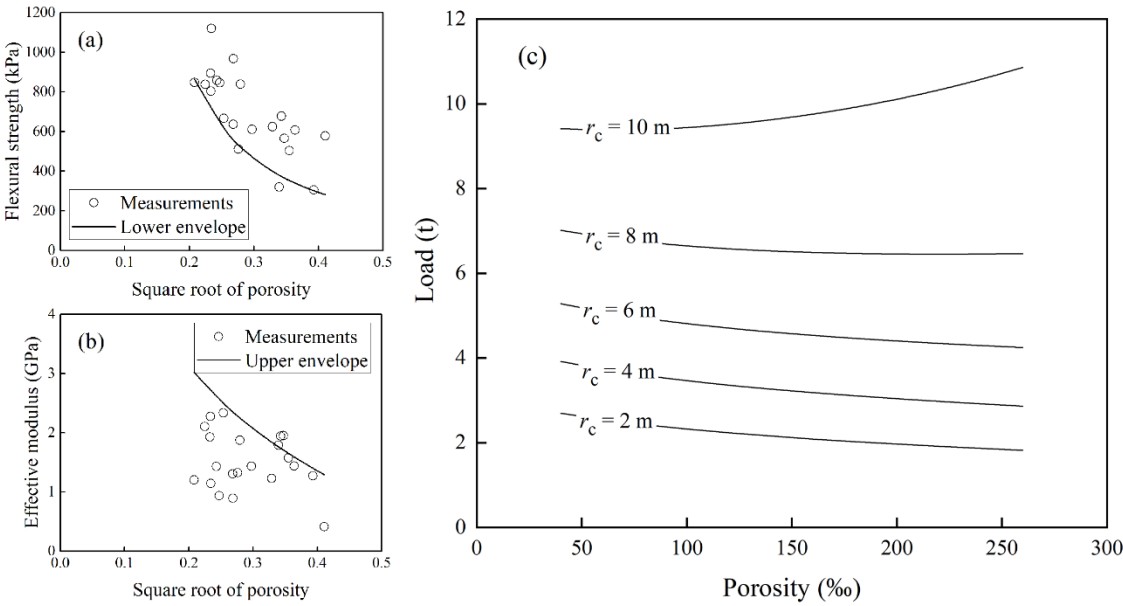

**Figure 14: Variations of load with respect to sea ice porosity for a landfast sea ice with congelation ice thickness of 1.3 m.**

It is emphasized that there are uncertainties in the estimated bearing capacity. First, sea ice mechanical properties were obtained using small-scale samples in this paper, which overestimates the full-scale cantilever beam tests preferred by industry standard. Additionally, the load calculated by Eq. (9) and (10) is for short term, under which the creep of ice is ignorable. Although the goods are transported away shortly after being unloaded on ice, the duration is at least hours because of manual operation, possibly resulting in ice creep. In the industry standard, in addition to ensure the ice sheet extreme fiber stress less than the allowable flexural stress for ice, it is necessary to avoid submergence caused by sea ice creep deformation (ISO19906, 2019). Therefore, the mass of goods should be less than the estimations given in Fig. 14c.

## 5 Conclusions

A series of laboratory mechanical experiments, including 25 bending and 55 uniaxial compression tests, was carried out on landfast sea ice collected in the Prydz Bay, East Antarctica. The flexural strength, effective modulus, vertically and horizontally loaded uniaxial compressive strength of congelation ice, snow ice and a mixture of the these two were measured. The strength of mixed ice is weaker than that of congelation ice because of the existence of snow ice, and that of ice consisting of pure snow ice is the lowest.

Sea ice mechanical properties are regional specific. The commonly used estimating equations of sea ice strength derived from North Hemisphere polar regions is not fit for landfast sea ice in the Prydz Bay, and alternative models are established for flexural and compressive strength (Eqs. 7 and 8). In addition to the uniaxial compressive strength that has already been related to sea ice porosity in previous researches, sea ice flexural strength is related to porosity in this work, rather than brine volume



that was often adopted in previous models. The newly proposed parameterization for flexural strength based on sea ice porosity compensates for the lack of applicability to warm ice using previous ones based on brine volume. Furthermore, the dependence

of sea ice strength on total porosity, rather than brine content can reflect the potential effects of climate change. Sea ice in polar regions may become warmer under the effects of global warming (Clem et al., 2020; Screen and Simmonds, 2010), and the gas within sea ice may occupy more space than brine (Wang et al., 2020).

As scientific investigations have flourished in the south pole region, the mechanical properties of Antarctic sea ice need to be urgently elucidated for safe activities on ice. With minimum flexural strength and maximum effective modulus of congelation

ice, a method is established to estimate the bearing capacity of landfast sea ice cover in the Prydz Pay following the industry standard (ISO19906, 2019). In this way, it is possible to estimate the magnitude of load that can be safely put on ice, based on the sea ice physical properties.

Admittedly, the mechanical tests performed in this paper is only derived from the extracted ice block, so the sample amount is limited. Nevertheless, the dataset provided here is of great value as it contains most industry-concerned sea ice mechanical

properties, which is helpful to improve the understanding of Antarctic sea ice and support of safe marine activities.




## Appendix A: Details of ice samples

A total of 25 bending and 55 uniaxial compression tests were performed on landfast sea ice samples. The detailed information of the ice samples is listed in Tables A1–A3, where C, S and M denote congelation ice, snow ice, and a mixture of the these
two, respectively.

**Table A1: Detailed information of bending test samples.**

| No. | Temperature (°C) | Porosity (‰) | Grain size (cm) | Platelet spacing (cm) | Flexural strength (kPa) | Effective modulus (GPa) | Ice type |
|-----|------|------|------|------|--------|-----|----|
| 1 | −12 | 54 | 0.84 | 0.13 | 894.4 | 1.9 | C |
| 2 | −12 | 72 | 0.54 | 0.09 | 636.5 | 1.3 | C |
| 3 | −12 | 72 | 0.94 | 0.09 | 966.9 | 0.9 | C |
| 4 | −12 | 51 | 0.87 | 0.10 | 837.5 | 2.1 | C |
| 5 | −8 | 55 | 0.62 | 0.18 | 804.1 | 2.3 | C |
| 6 | −8 | 115 | 0.94 | 0.11 | 319.5 | 1.8 | C |
| 7 | −8 | 132 | 0.73 | 0.09 | 608.3 | 1.4 | C |
| 8 | −8 | 126 | 0.65 | 0.09 | 503.6 | 1.6 | C |
| 9 | −5 | 55 | 0.67 | 0.11 | 1119.7 | 1.1 | C |
| 10 | −5 | 43 | 0.82 | 0.08 | 847.9 | 1.2 | C |
| 11 | −5 | 64 | 0.90 | 0.11 | 665.8 | 2.3 | C |
| 12 | −5 | 59 | 0.58 | 0.10 | 858.0 | 1.4 | C |
| 13 | −3 | 169 | 1.05 | 0.07 | 577.8 | 0.4 | C |
| 14 | −3 | 154 | 1.03 | 0.07 | 305.3 | 1.3 | C |
| 15 | −3 | 108 | 0.86 | 0.09 | 624.1 | 1.2 | C |
| 16 | −3 | 88 | 1.25 | 0.07 | 611.1 | 1.4 | C |
| 17 | −3 | 121 | / | / | 566.0 | 2.0 | C |
| 18 | −5 | 76 | / | / | 511.3 | 1.3 | C |
| 19 | −5 | 78 | / | / | 838.8 | 1.9 | C |
| 20 | −5 | 61 | / | / | 845.9 | 0.9 | C |
| 21 | −5 | 117 | / | / | 677.6 | 1.9 | C |
| 22 | −3 | / | / | / | 105.8 | 0.4 | S |
| 23 | −3 | / | / | / | 92.9 | 0.2 | S |
| 24 | −3 | / | / | / | 115.1 | 0.5 | S |
| 25 | −5 | / | / | / | 176.9 | 0.4 | S |



**Table A2: Detailed information of vertically loaded uniaxial compressive test samples at the temperature of −3ºC.**

| No. | Strain rate (s⁻¹) | Porosity (‰) | Uniaxial compressive strength (MPa) | Failure mode | Ice type |
|---|---|---|---|---|---|
| 1 | $1.1 \times 10^{-6}$ | 120 | 1.8 | Ductile | C |
| 2 | $1.9 \times 10^{-6}$ | 102 | 2.1 | Ductile | C |
| 3 | $5.9 \times 10^{-6}$ | 92 | 2.4 | Ductile | C |
| 4 | $1.4 \times 10^{-5}$ | 96 | 3.5 | Ductile | C |
| 5 | $2.9 \times 10^{-5}$ | 86 | 4.9 | Ductile | C |
| 6 | $3.0 \times 10^{-5}$ | 106 | 4.4 | Ductile | C |
| 7 | $4.3 \times 10^{-5}$ | 99 | 3.1 | Ductile | C |
| 8 | $6.0 \times 10^{-5}$ | 128 | 3.5 | Ductile | C |
| 9 | $8.7 \times 10^{-5}$ | 145 | 2.7 | Ductile | C |
| 10 | $8.9 \times 10^{-5}$ | 111 | 3.1 | Ductile | C |
| 11 | $9.6 \times 10^{-5}$ | 99 | 3.4 | Ductile | C |
| 12 | $1.4 \times 10^{-4}$ | 113 | 4.6 | Ductile | C |
| 13 | $1.9 \times 10^{-4}$ | 63 | 6.0 | Ductile | C |
| 14 | $3.0 \times 10^{-4}$ | 82 | 5.2 | Ductile | C |
| 15 | $5.7 \times 10^{-4}$ | 102 | 6.7 | Brittle | C |
| 16 | $9.1 \times 10^{-4}$ | 120 | 3.7 | Brittle | C |
| 17 | $3.0 \times 10^{-3}$ | 62 | 4.5 | Brittle | C |
| 18 | $6.0 \times 10^{-3}$ | 97 | 3.9 | Brittle | C |
| 19 | $9.0 \times 10^{-3}$ | 166 | 2.4 | Brittle | C |
| 20 | $2.0 \times 10^{-4}$ | / | 1.5 | Creep in snow ice | M |
| 21 | $2.0 \times 10^{-4}$ | / | 1.8 | Creep in snow ice | M |
| 22 | $2.0 \times 10^{-3}$ | / | 1.1 | Brittle | M |
| 23 | $2.0 \times 10^{-3}$ | / | 1.0 | Brittle | M |
| 24 | $1.0 \times 10^{-4}$ | / | 0.4 | Creep | S |
| 25 | $2.0 \times 10^{-4}$ | / | 0.3 | Creep | S |
| 26 | $3.5 \times 10^{-4}$ | / | 0.4 | Creep | S |
| 27 | $6.0 \times 10^{-4}$ | / | 0.2 | Brittle | S |
| 28 | $2.0 \times 10^{-3}$ | / | 0.1 | Brittle | S |






**Table A3: Detailed information of horizontally loaded uniaxial compressive test samples at the temperature of −3°C.**

| No. | Strain rate (s$^{-1}$) | Porosity (‰) | Uniaxial compressive strength (MPa) | Failure mode | Ice type |
|---|---|---|---|---|---|
| 1 | $1.0 \times 10^{-5}$ | 82 | 0.7 | Ductile | C |
| 2 | $1.0 \times 10^{-5}$ | 143 | 0.7 | Ductile | C |
| 3 | $2.0 \times 10^{-5}$ | 73 | 0.6 | Ductile | C |
| 4 | $3.0 \times 10^{-5}$ | 114 | 0.8 | Ductile | C |
| 5 | $6.0 \times 10^{-5}$ | 118 | 0.9 | Ductile | C |
| 6 | $1.0 \times 10^{-4}$ | 124 | 0.8 | Ductile | C |
| 7 | $1.4 \times 10^{-4}$ | 102 | 1.3 | Ductile | C |
| 8 | $2.0 \times 10^{-4}$ | 71 | 1.6 | Ductile | C |
| 9 | $3.5 \times 10^{-4}$ | 65 | 2.2 | Ductile | C |
| 10 | $3.5 \times 10^{-4}$ | 97 | 1.4 | Ductile | C |
| 11 | $6.0 \times 10^{-4}$ | 94 | 1.1 | Ductile | C |
| 12 | $6.0 \times 10^{-4}$ | 140 | 1.7 | Ductile | C |
| 13 | $1.0 \times 10^{-3}$ | 97 | 2.0 | Ductile | C |
| 14 | $1.4 \times 10^{-3}$ | 95 | 2.2 | Ductile | C |
| 15 | $1.8 \times 10^{-3}$ | 113 | 2.1 | Ductile | C |
| 16 | $2.0 \times 10^{-3}$ | 118 | 1.9 | Brittle | C |
| 17 | $2.7 \times 10^{-3}$ | 80 | 2.3 | Brittle | C |
| 18 | $3.5 \times 10^{-3}$ | 114 | 1.9 | Brittle | C |
| 19 | $6.0 \times 10^{-3}$ | 95 | 1.6 | Brittle | C |
| 20 | $1.0 \times 10^{-2}$ | 94 | 1.2 | Brittle | C |
| 21 | $1.0 \times 10^{-4}$ | / | 0.3 | Brittle | S |
| 22 | $2.0 \times 10^{-4}$ | / | 0.3 | Brittle | S |
| 23 | $3.5 \times 10^{-4}$ | / | 0.2 | Brittle | S |
| 24 | $6.0 \times 10^{-4}$ | / | 0.2 | Brittle | S |
| 25 | $2.0 \times 10^{-3}$ | / | 0.1 | Creep | S |
| 26 | $3.5 \times 10^{-3}$ | / | 0.1 | Creep | S |
| 27 | $6.0 \times 10^{-3}$ | / | 0.1 | Creep | S |



**Data availability**

All data are available at the website (https://doi.org/10.5281/zenodo.5787915).

**Author contribution**

QW, YX, and ZJL planned the experiments; QW and ZQL performed the experiments; QW analyzed the data; QW wrote the manuscript draft; PL reviewed and edited the manuscript.

**Competing interests**

The authors declare that they have no conflict of interest.

**Acknowledgement**

This research was supported by the National Natural Science Foundation of China (52192692, 41906198 and 41922045), the Fundamental Research Funds for the Central Universities (DUT21RC3086), and the Liao Ning Revitalization Talents Program

(XLYC2007033 and XLYC1908027). We thank the crews of PRIC for their assistance with the fieldwork during ice sampling, and Shiping Liu from the Xinhua News Agency for providing the photo of unloading cargos on ice.

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
