# Peer review of "Flexural and compressive strength of the landfast sea ice in the Prydz Bay, East Antarctic"

_The Cryosphere, 2021_

## Referee Comment (RC2)

**Flexural and compressive strength of the landfast sea ice in the Prydz Bay, East Antarctica**
**by Wang and others**

**General comments**

I suggest you try to keep the result to *your own results only*. The comparison to others and discussion on why fit better in the Discussion section. For example sub-section 3.2.1 is almost only comparison with others and discussions on why. Put this content in the Discussion section.

**1. Introduction**

OK, perhaps also refer to Strub-Klein and Høyland (2012).

**2. In-situ sampling and laboratory experiments**

**2.1. In situ sampling**

1. Ice temperature profile during field work? I suggest you move this information from section 4.2 into the *In-situ sampling* section.

2. What was the air temperature during field work? Do you have a air temperature history a few weeks back?

3. How long time did the field work take? Or how long was the ice exposed to the air temperatures and possibly solar radiation?

**2.2. Laboratory experiments**

**2.3. Crystal structure**

**2.4. Bending tests**

Elastic modulus. Could you explain how you derived these? Equation 2 only give a force and a displacement. But, there must be some kind of $\Delta F / \Delta \delta$? There are several ways to do this, one may search for the steepest part of the curve, use some kind of average etc.

**2.5. Compression tests**

1. Measure of displacement. I assume this is the position of the loaded plate and not the compression of the ice sample? I don't know your machine, but usually there is some elasticity in the machine that gives a somewhat lower compression of the sample than what is given by the displacement of the loaded plate.

2. Equation 3. Perhaps use $F_{max}$?

**2.6. Uncertainty analysis**

The numbers here could be used to give a reasonable amount of numbers in the dervied properties.

**3. Results**

**3.1. Crystal structure**

Where is the water line in Figure 4?

**3.2. Flexural strength**

1. The values of flexural strength are given with a lot of number. But, if you consider an uncertainty of 0.002 and a value of about 700 it should be sufficient to give numbers like 511 kPA, 846 kPa etc.

2. Why not give flexural strength of snow ice also here?

3. As explained above I suggest to move the content of sub-section 3.2.1 (Congelation ice) to Discussion.

4. Line 192. *the region specific*. I don't like this explanation at all. The ice does not know where it is, it only knows which physical conditions it has been exposed to. It is OK if you cannot explain why things are different, but do not blindly blame Geography!

5. Differences to Timco and O'Brien (1994). T&B give some kind of upper limit and this means that almost any set of experiments will give lower average values. In other words it is natural that you find lower values.

6. Differences to Karulina et al. (2019). Here your results are higher and there are some obvious differences that should be discussed. Firstly, Karulina et al. (2019) tested in field, secondly they tested larger beams larger beams. It could be that their beams had more weaknesses than yours. You prepared the beams carefully in the lab and these two facts may help to explain. Also the different testing methods may have contributed.

7. It is interesting and new that you investigate the flexural strength in relation to grain size and platelet spacing. Very nice.

8. Figure 5. It is interesting to note that the slope was more or less equal for the columnar and mixed ice, in spite of different strengths. And that the peak deformation was equal for the snow ice and the mixed ice in spite of very different strengths. Was this coincidental?

**3.3. Effective modulus**

1. As explained above you need to explain how you found the effective elastic modulus ($E$).

2. $E$ is a function of force, displacement and time. The more time a tests takes the more important becomes the viscous (or delayed viscous) deformation. The time-dependent deformation is know to be a function of salinity(brine volume). Did Karulina et al. (2019) load with the same load/displacement rate as you did? If they loaded more slowly it may explain why they found $E = f(brinevolume)$?

**3.4. Uniaxial compressive strength**

**3.4.1. Congelation ice**

There is much more available published data on uni-axial strength and it is good to see that your results are more or less in line with what we think we know from before.

**3.4.2. Mixed and snow ice**

Any comment on physical properties of the snow ice? You do not report densities or porosities. Why? If the ice was too porous to shape samples properly, please say so. Did you have any impression from visual observation? Was the ice more porous or why was it weaker?

**4. Discussion**

**4.1. Ratios between strengths**

You could also compare with Moslet (2007) and Strub-Klein and Høyland (2012), they also report vertical / horizontal uni-axial compression strengths. I don't think you can claim that you have found the unique ratios between uni-axial compression in vertical direction, the same in horizontal direction and flexural strengths. Moslet (2007) argues that this is a function of ice temperature among other things.

**4.2. Comparison between field and lab**

This discussion should be linked to the comparison with Karulina et al. (2019). One important aspect I suggest you think about is cooling and then heating of the sea ice. We have tested relatively warm ice in-situ, then sampled cooled down (-15C) and stored (some weeks or some months), and finally heated again and tested. The samples that were cooled down and heated again were clearly stronger than the in-situ ice even if the temperature was the same! I think this is an important, and not understand mechanism in ice mechanics that should be studied, it may explain why SYI and Old Ice are both stronger than FYI even for comparable temperatures, and porosities.

**4.3. Bearing capacity of landfast sea ice in the Prydz Bay**

**5. Conclusions**

**References**

Karulina, M., Marchenko, A., Karulin, E., Sodhi, D., Sakharov, A., Chistyakov, P., 2019. Full-scale flexural strength of sea ice and freshwater ice in Spitsbergen Fjords and North-West Barents Sea. Applied Ocean Research (90, 101853), https://doi.org/10.1016/j.apor.2019.101853.

Moslet, P. O., 2007. Field testing of uniaxial compression strength of columnar sea ice. Cold Regions Science and Technology (48 (1)), 1-14.

Strub-Klein, L., Høyland, K. V., 2012. Spatial and temporal distributions of level ice properties: Experimental and thermo-mechanical analysis. Cold Regions Science and Technology 71, 11-22.

Timco, G. W., O'Brien, S., 1994. Flexural strength equations for sea ice. Cold Regions Science and Technology (22), 285-298.

---

## Author Comment (AC1)

**Response to the Comments of Reviewer1**

This paper presents mechanical property test results of Antarctic sea ice and links those to the prevailing physical properties including porosity, brine volume, grain size, platelet spacing and strain rate. The paper contributes to the state of the art by providing valuable insights of the applicability of several existing methods to the estimation of Antarctic sea ice properties, specifically in the Prydz Bay, and by offering location-specific ice mechanical property and bearing capacity estimation for engineering purposes. The extensive effort to accomplish the research purpose is appreciated and the results are presented and analysed in a logical and clear manner.

We appreciate warmly for the reviewer's earnest work. The comments are constructive, and we have revised the manuscript accordingly. Detailed answers to all comments are provided below.

The specific comments are:

**Comment:** The brine volume and porosity were calculated using ice temperature, salinity and density using Cox and Weeks formulae. The calculation will most likely involve uncertainties which may have an impact on the later investigations. The authors are suggested to comment on the significance of this uncertainty source and its influence on the results of this work.

**Response:** Thanks. We will stress this issue in the revised manuscript. Since it is not easy to quantify the uncertainties (the error propagation estimation needs independent direct measured variables, see respond below), so we talk about it in a qualitative way as below.

It is noteworthy that the calculation most likely involves uncertainties introduced by the measurement errors of ice physical properties, especially for sea ice porosity, of which the air volume fraction is largely dependent on ice density (Timco and Frederking, 1996).

**Comment:** Line 105: the authors are suggested to specify the speed of loading. It is not very clear what 'time-of-loading' means. I assume the ice beam fails very soon after loaded.

**Response:** We will use strain rate to define loading speed in the new version. The strain

rate in three-point bending test is calculated using equation below (see Han et al. (2016))

$$\dot{\varepsilon}_\mathrm{f} = \frac{6h\dot{\delta}}{l^2}$$

where $\dot{\varepsilon}_\mathrm{f}$ is strain rate of bending test; $\dot{\delta}$ is displacement rate of the pressing plate; $l$ is span between supports; $h$ is height of the beam. Result shows that the strain rate of our bending tests varies from $10^{-5}$ to $10^{-3}$ s$^{-1}$.

**Comment:** Line 199-200: some example references can be added to explain 'other commonly used functions'

**Response:** The reported relationships between sea ice flexural strength and square root of brine volume fraction were in exponential (Timco and O'Brien, 1994) and linear froms (Krupina and Kubyshkin, 2007). In this paper, we adopted more expressions including exponential, linear, logarithm and power functions. We will list these mathematical functions we used as we think it may be much clear than showing the references.

**Comment:** The confidence intervals adopted for various analyse vary from 90% (e.g. Figure 7) to 99% (Figure 6). Is there a ration behind the selection of confidence intervals?

**Response:** The confidence intervals are determined according to the individual significance levels ($p$) obtained by regression analyses. For example, in Fig. 6, $p$ of the best-fit relationship between flexural strength and square root of porosity was less than 0.01, so we chose 99% as the confidence interval. In Fig. 7, $p > 0.1$ for the flexural strength-grain size best-fit equation, so the confidence interval was selected as 90%; and $p < 0.05$ for the flexural strength-platelet spacing best-fit equation, so the confidence interval was selected as 95%. Moreover, for the best-fit equations with various significant levels in different regimes, such as in Fig. 10a, we chose the maximum value as the final confidence interval for all the best-fit lines.

**Comment:** It would be helpful to indicate the range of salinity measured among the samples. It is found that the flexural strength is not sensitive to brine volume. Would it be possible that this is because of the small range of salinity coverd by the samples (since they are from the same ice block)?

**Response:** Yes, what the reviewer suggested could be a reason. The salinity of congelation-ice samples in the bending tests was 1.0–5.1 psu. The brine volume fraction is a function of ice temperature, salinity, and density, and the square root of brine

volume fraction of our samples was 0.11–0.27. The range is narrower than that reported in Timco and O'Brien (1994) (0–0.5) and in Karulina et al. (2019) (0.16–0.39), which probably makes that the flexural strength of our samples is not sensitive to brine volume. We will also add the above discussion in the revised manuscript.

**Comment:** How does Eq. (7) compare to the existing equations in the literature? Are they similar or do they differ a lot?

**Response:**

(1) First, as shown in the Figure (a) below, the flexural strength estimated using Eq. (7) agrees well with the measured strength with correlation coefficient of 0.75 ($p <$ 0.01).

(2) Second, to better compare our best-fit equation (Eq. 7) with existing equations reported in Karulina et al. (2019) and Timco and O'Brien (1994), the results of flexural strength calculated based on our test data were plotted against the square root of brine volume fraction in Figure (b). Results showed that the strength estimated using Karulina et al. (2019) was much lower than that estimated using ours. The strength estimated using Timco and O'Brien (1994) agreed better with that estimated using ours than Karulina et al. (2019) and only overestimated by 1.1 times.

The above comparisons will be added in the revised manuscript.

[Figure]

(a)  (b)

**Figure** Comparisons of estimated strength using our best-fit equation with (a) measured strength and (b) estimated strength using existing equations in the literature.

**Comment:** Line 258-259: the sample size may be too small to draw the conclusion on temperature effect.

**Response:** The statement about the effect of ice temperature on snow-ice effective modulus will be deleted.

**Comment:** The first paragraph of 3.4.1: nice and thorough explanations are provided here to explain the measured trend of compressive strengths. More references are suggested here to support the reasoning, so that it does not look like own speculation. Same for later parts with such explanations.

**Response:** Thanks. More references of Gold (1997), Ji et al. (2020), Kuehn and Schulson (1994), Sinha (1988), and Schulson (2001) will be cited to support the relative statements in this section.

**Comment:** The small size ice samples are cut from different positions along the thickness direction. Does the measured mechanical properties exhibit dependence on the thickness position? Typically congelation columnar ice is stronger at the top than at the bottom. This relates to Figure 14, where all the measurement has been plotted together in the same figure. The lower evelope probably corresponds mainly to flexural strength at the bottom, while in the case of bearing capacity ice fails at the top layer. This leads to conservative estimation of the bearing capacity.

**Response:**

(1) Due to the limited number of samples under each ice temperature and the focus of examining the effects of porosity and brine volume on sea ice strength, we did not record the thickness position of our bending samples in the whole ice sheet. Therefore, the dependence of strength on the ice depth is not able to be checked here. In general, as the reviewer said, the ice is stronger at the top than at the bottom.

(2) For estimating the bearing capacity of landfast sea ice, as the reviewer said, we conducted a conservative estimation. Because the real scenario is that the cargos are unloaded on the ice sheet, and thus, the strength of ice sheet is needed rather than that of small-scale samples. While the elastic modulus of sea ice varied along ice thickness, making it difficult to obtain the real distribution of stress along ice thickness. So, we conducted a conservative method for safe designing in this paper by adopting the minimum flexural strength. All the measured strength of ice samples was plotted in Fig. 14a, and the lower envelope of flexural strength was selected to represent the strength of ice sheet. The results indicate a minimum load that can put on ice.

In addition, we think that the above assumption is close to the actual scenario to some degree. As the load is applied on ice sheet, the sheet should be compressed at the top and tensioned at the bottom. Ice is a material which is strong in compression and weak in tension. So, the ice sheet deflects until the first crack or yielding

develops in the underside of the sheet beneath the center of the load (Masterson, 2009). The low strength often occurs at the bottom of ice sheet because of the higher ice temperature near freezing point; therefore, it is reasonable to use the lower envelope of flexural strength.

The above discussion will be added in the revised manuscript.

**Comment:** It may be worth also mentioning the influence of platelet spacing in the conclusion part.

**Response:** Thanks. The statement below will be added in *Section Conclusions*:

The effects of sea ice sub-structure on columnar ice strength were investigated. Both flexural strength and effective elastic modulus increase with increasing platelet spacing, while the influence of grain size is not significant.

Some technical corrections:

**Comment:** Line 51: the statement after 'because' tells why there are more understanding of mechanical properties of Arctic sea ice, but not really the reason why there are very few for the Antarctic. Consider rephrasing to make it more natural.

**Response:** The statement will be rephrased as below:

The mechanical properties of Arctic sea ice have been widely investigated in the last century because of booming oil and gas exploration in the Northern Hemisphere polar regions. While the understanding of mechanical properties of Antarctic sea ice is limited due to less human and industry activities than those developed in the Arctic.

**Comment:** Line 53: 'south pole' means exactly the pole (latitude 90). Here it should be something like 'Antarctic continent'.

**Response:** It will be replaced with Antarctic regions.

**Comment:** Line 128: rule -> ruler?

**Response:** Yes, it is ruler. Corrected accordingly.

**Comment:** Figure 4b: the pictures are small, making it difficult to see clearly the crystal structures. Consider enlarging.

**Response:** A much clearer figure will be exhibited as below.

[Figure]

**Comment:** Eq. (8): typically equation follows immediately where it is firstly mentioned -> move 'overestimation ...' to after Eq. (8)

**Response:** Corrected accordingly.

**Comment:** Line 379: empirical -> empirically

**Response:** Corrected accordingly.

**Comment:** Line 420: photted -> photoed

**Response:** Corrected accordingly.

**Comment:** Line 438: radiuses -> radii

**Response:** Corrected accordingly.

Reference

Gold, L.: Statistical characteristics for the type and length of deformation-induced cracks in columnar-grain ice, J. Glaciol., 43, 311–320, https://doi.org/10.3189/S0022143000003269, 1997.

Han, H., Jia, Q., Huang, W., and Li, Z.: Flexural strength and effective modulus of large columnar-grained freshwater ice, J. Cold Reg. Eng., 30, 04015005, https://doi.org/10.1061/(ASCE)CR.1943-5495.0000098, 2016.

Ji, S., Chen, X., and Wang, A.: Influence of the loading direction on the uniaxial compressive strength of sea ice based on field measurements, Ann. Glaciol., 61, 86–96, https://doi.org/10.1017/aog.2020.14, 2020.

Karulina, M., Marchenko, A., Karulin, E., Sodhi, D., Sakharov, A., and Chistyakov, P.:
Full-scale flexural strength of sea ice and freshwater ice in Spitsbergen Fjords and
North-West Barents Sea, Appl. Ocean Res., 90, 101853,
https://doi.org/10.1016/j.apor.2019.101853, 2019.

Krupina, N. A. and Kubyshkin, N. V.: Flexural strength of drifting level first-year ice
in the Barents Sea, in: Proceedings of the 17th International Offshore and Polar
Engineering Conference, Portugal, 1–6 July 2007, 2007

Kuehn, G. and Schulson, E.: The mechanical properties of saline ice under uniaxial
compression, Ann. Glaciol., 19, 39–48, https://doi.org/10.3189/1994AoG19-1-39-
48, 1994.

Masterson, D. M.: State of the art of ice bearing capacity and ice construction, Cold
Reg. Sci. Technol., 58, 99–112, https://doi.org/10.1016/j.coldregions.2009.04.002,
2009.

Sinha, N. K.: Crack-enhanced creep in polycrystalline material: strain-rate sensitive
strength and deformation of ice, J. Mater. Sci., 23, 4415–4428,
https://doi.org/10.1007/BF00551940, 1988.

Schulson, E. M.: Brittle failure of ice, Eng. Fract. Mech., 68, 1839–1887,
https://doi.org/10.1016/S0013-7944(01)00037-6, 2001.

Timco, G. W. and Frederking, R. M. W.: A review of sea ice density, Cold Reg. Sci.
Technol., 24, 1–6, https://doi.org/10.1016/0165-232X(95)00007-X, 1996.

Timco, G. W. and O'Brien, S.: Flexural strength equation for sea ice, Cold Reg. Sci.
Technol., 22, 285–298, https://doi.org/10.1016/0165-232X(94)90006-X, 1994.

---

## Author Comment (AC2)

**Response to the Comments of Reviewer2**

An interesting paper, congratulations with all the good field work.

**Response:** We thank the reviewer for his recognition of our work. The comments are detailed and constructive, based on which we have revised the manuscript carefully. Please find our responses to individual comments below.

**General comments**

**Comment:** I suggest you try to keep the result to *your own results only*. The comparison to others and discussion on why fit better in the Discussion section. For example subsection 3.2.1 is almost only comparison with others and discussions on why. Put this content in the Discussion section.

**Response:** Thanks. A new section 4.1 Comparisons with previous studies will be added in the Discussion section, and the comparisons to other studies on flexural strength, effective modulus, and compressive strength are to be moved to this section.

**1. Introduction**

**Comment**: OK, perhaps also refer to Strub-Klein and Høyland (2012). **Response**: Their work will be cited.

**2. In-situ sampling and laboratory experiments**

**2.1. In situ sampling**

**Comment**: Ice temperature profile during field work? I suggest you move this information from section 4.2 into the *In-situ sampling* section.

**Response**: Corrected accordingly.

**Comment**: What was the air temperature during field work? Do you have a air temperature history a few weeks back?

**Response:**

- (1) During the field tests, the air temperature varied from -2.6 to  $1.8^{\circ}$ C with an average of  $-0.8\pm0.9^{\circ}$ C. We will add the information in the new version.
- (2) There is a weather station at the Zhongshan station. Since the field work site was not far away the Zhongshan station, so the air temperature recorded by the weather station is used. The figure below shows the air temperature in the two months

before field work. A rise in the air temperature occurred after 15 October 2019 (UTC) from below  $-10^{\circ}$ C to above  $-10^{\circ}$ C. This information and figure will also be added.

Figure The air temperature from October 1 to November 24 2019 at Zhongshan station

**Comment**: How long time did the field work take? Or how long was the ice exposed to the air temperatures and possibly solar radiation?

**Response**: Approximately 2 hours after lifting onto the deck, part of the ice block was cut and machined into samples, and the bending tests were completed. During the tests, the air temperature varied from -2.6 to  $1.8^{\circ}$ C with an average of  $-0.8\pm0.9^{\circ}$ C, and it was overcast with low solar radiation. We will add the above information in the revised manuscript.

**2.4. Bending tests**

Comment: Elastic modulus. Could you explain how you derived these? Equation 2 only give a force and a displacement. But, there must be some kind of  $\Delta F/\Delta \delta$ ? There are several ways to do this, one may search for the steepest part of the curve, use some kind of average etc.

**Response:** If the load is applied on the midspan of a simply supported beam, according to simple elastic beam theory, the midspan deflection of beam is

$$\delta = \frac{Fl^3}{4bh^3E}$$

where  $\delta$  is midspan deflection, F is force at failure, E is Elastic modulus, l is span between supports, b and h are section width and height of the beam.

In the ice bending tests, with an assumption that the beam is perfectly elastic, the Eleatic modulus can be then derived using Eq. (2) in our paper if  $\delta$  is known. The equation has

been suggested by IAHR Section on Ice Problems (Schwarz et al., 1981) and adopted by other reports (Karulina et al., 2019; Kermani et al., 2008). As sea ice is not turly elastic, and the derived modulus is termed effective modulus (Timco and Frederking, 2010). We will give a much clearer explanation on the equation.

Additionally, combined with the third reviewer's comment, the term of E will be changed to effective elastic modulus.

**2.5. Compression tests**

**Comment:** Measure of displacement. I assume this is the position of the loaded plate and not the compression of the ice sample? I don't know your machine, but usually there is some elasticity in the machine that gives a somewhat lower compression of the sample than what is given by the displacement of the loaded plate.

**Response:** Yes, the measured displacement is the position of the loading plate. The deformation of the test machine results in a lower compression of the ice sample than the displacement of the loaded plate. Therefore, the true stain rate of the ice sample is less than the nominal strain rate of the test (Timco and Frederking, 1984), and the latter was used in this paper. We used a universal testing machine to measure sea ice uniaxial compressive strength. The accurate stiffness of the machine is not measured. The machine is equipped with a portal frame with four columns supporting the upper beams and the working plate, all of which are made of welded steel plates. So, it is expected to be rigid enough and produce a minor effect on the compressive tests.

More detailed information on our test machine and the effect of stiffness on strain rate will be added in the revised manuscript.

**Comment**: Equation 3. Perhaps use *Fmax*? **Response**: Corrected accordingly.

**2.6. Uncertainty analysis**

**Comment**: The numbers here could be used to give a reasonable amount of numbers in the dervied properties.

**Response:** This section will be deleted in the new manuscript because we found that we had missed the precondition of the error propagation that the direct measured variables used to derive indirect variables must be independent. While the force applied and sample dimensions are actually correlated.

**3. Results**

**3.1. Crystal structure**

**Comment**: Where is the water line in Figure 4?

**Response:** The accurate ice freeboard was not measured in field as our focus was paid on snow and ice thickness. The freeboard was derived from a video recording the ice side surface by comparing with the snow thickness, and it was 19 cm approximately. We will add the information in text and figure.

**3.2. Flexural strength**

**Comment**: The values of flexural strength are given with a lot of number. But, if you consider an uncertainty of 0.002 and a value of about 700 it should be sufficient to give numbers like 511 kPA, 846 kPa etc.

**Response:** As the respond to previous comment, the original uncertainty analysis will be deleted due to wrong understanding. While we agree the reviewer that the strength value with integral number is enough considering the accuracy of force sensor and caliper. Therefore, we will correct the values of flexural strength.

**Comment**: Why not give flexural strength of snow ice also here? **Response**: We will give this information in the revised manuscript.

**Comment**: As explained above I suggest to move the content of sub-section 3.2.1 (Congelation ice) to Discussion.

**Response:** Corrected accordingly.

**Comment:** Line 192. *the region specific*. I don't like this explanation at all. The ice does not know where it is, it only knows which physical conditions it has been exposed to. It is OK if you cannot explain why things are different, but do not blindly blame Geography!

**Response:** Thanks for your comment. We will delete the original explanation and rephrase it according to your comment below.

**Comment**: Differences to Timco and O'Brien (1994). T&B give some kind of upper limit and this means that almost any set of experiments will give lower average values. In other words it is natural that you find lower values.

**Response:** Discussion will be added following your suggestion in the revised

**manuscript.**

**Comment**: Differences to Karulina et al. (2019). Here your results are higher and there are some obvious differences that should be discussed. Firstly, Karulina et al. (2019) tested in field, secondly they tested larger beams larger beams. It could be that their beams had more weaknesses than yours. You prepared the beams carefully in the lab and these two facts may help to explain. Also the different testing methods may have contributed.

**Response:** We will revise the discussion based on your comments as below.

The differences may be attributed to several facts. The first is that our tests were conducted in the laboratory while the tests in Karulina et al. (2019) were performed in field, so that our samples were able to be prepared in caution. On the other hand, the ice samples in Karulina et al. (2019) were much larger and contained more potential weaknesses than ours. Besides, the flexural strength in Karulina et al. (2019) was derived using cantilever beams, and stress concentrations at the root of beam resulted in low strength.

**Comment**: It is interesting and new that you investigate the flexural strength in relation to grain size and platelet spacing. Very nice.

**Response**: Thanks for the reviewer's approval on this work.

**Comment**: Figure 5. It is interesting to note that the slope was more or less equal for the columnar and mixed ice, in spite of different strengths. And that the peak deformation was equal for the snow ice and the mixed ice in spite of very different strengths. Was this coincidental?

**Response:** We have checked all our data and found that:

- The mean slopes of force varying with deformation are similar between columnar ice (512±246 N⋅mm-1) and mixed ice (625±178 N⋅mm-1), and both of them were higher than snow ice (157±46 N⋅mm-1)
- (2) The mean peak deflections of samples at failure are similar between snow ice (0.31±0.11 mm) and mixed ice (0.41±0.21 mm).

We will add the above information in the revised manuscript.

3.3. Effective modulus

Comment: As explained above you need to explain how you found the effective elastic modulus (E).

**Response**: Thanks, please see the above respond where we have made a detailed explanation.

**Comment**: *E* is a function of force, displacement and time. The more time a tests takes the more important becomes the viscous (or delayed viscous) deformation. The time-dependent deformation is know to be a function of salinity(brine volume). Did Karulina et al. (2019) load with the same load/displacement rate as you did? If they loaded more slowly it may explain why they found E = f (brinevolume)?

**Response:**

- (1) Based on the first reviewer's comment, we have calculated the strain rate of our bending tests, and stain rate ranged between  $10^{-5}$  to  $10^{-3}$  s-1.
- (2) The strain rate in Karulina et al. (2019) varied from  $10^{-4}$  to  $10^{-3}$  s-1. So, our tests were performed at a similar or even a bit slower rate than Karulina et al. (2019). The time-dependent deformation seems not the reason that *E* is the function of brine volume in Karulina et al. (2019) while not in this paper. We will also add the above discussion in the revised manuscript.
- 3.4. Uniaxial compressive strength
- 3.4.1. Congelation ice

**Comment**: There is much more available published data on uni-axial strength and it is good to see that your results are more or less in line with what we think we know from before.

**Response:** Yes, many previously published studies have reported similar results as given in our paper. In the revised manuscript, we will cite more references and give a brief statement on these similarities.

- (1) For stress-strain curves, Bonath et al. (2019) and Arakawa and Maeno (1997) are commented and cited.
- (2) For the effect of sea ice porosity on compressive strength, Moslet (2007) is commented and cited.
- (3) For the failure modes compressed under different directions, Gold (1997), Kuehn and Schulson (1994), and Sinha (1988) are cited.
- (4) For the effect of stain rate on compressive strength and corresponding fitting

equations, Arakawa and Maeno (1997), Schulson (2001), Timco and Frederking (1990), and Høyland (2007) are cited.

(5) For the comparisons between horizontally- and vertically loaded strength, Strub-Klein and Høyland (2012) is cited.

**3.4.2. Mixed and snow ice**

**Comment:** Any comment on physical properties of the snow ice? You do not report densities or porosities. Why? If the ice was too porous to shape samples properly, please say so. Did you have any impression from visual observation? Was the ice more porous or why was it weaker?

**Response:**

- (1) The mean density of snow-ice samples in bending test was 0.55±0.01 g⋅cm-3, and that in compression test was 0.61±0.13 g⋅cm-3. The snow-ice samples could be distinguished easily by their white appearance and light weight.
- (2) As snow ice was not formed by congelation of sea water, the equations proposed by Cox and Weeks (1983) were not able to be used to determine porosity. From visual observation, the snow ice was much porous than the underlying congelation ice. The snow ice was compacted than new snow on top; therefore, snow ice can still be machined into regular shape.

We will add the above information in the revised manuscript.

**4. Discussion**

**4.1. Ratios between strengths**

**Comment:** You could also compare with Moslet (2007) and Strub-Klein and Høyland (2012), they also report vertical / horizontal uni-axial compression strengths. I don't think you can claim that you have found the unique ratios between uni-axial compression in vertical direction, the same in horizontal direction and flexural strengths. Moslet (2007) argues that this is a function of ice temperature among other things.

**Response:** We will compare our ratios with those reported in Moslet (2007) and Strub-Klein and Høyland (2012) in the revised manuscript. It was found that the ratio of vertically to horizontally loaded uniaxial compressive strength in our tests was independent of porosity, and the average was  $3.1\pm0.9$ . The ratio of vertically loaded uniaxial compressive strength to flexural strength decreased with increasing porosity, and it reached 8.0 for sea ice with small porosity and 4.0 for sea ice with large porosity approximately. The average ratio was  $7.4\pm1.9$ . Moslet (2007) reported a ratio between vertically and horizontally loaded strength of columnar ice of 1.3 for cold ice ( $<-10^{\circ}$ C) and of 4–5 for warm ice. Strub-Klein and Høyland (2012) reported low vertical-to-horizontal strength ratios of 1.4–1.8 for granular and columnar ice because the sampling ice cover had been broken and recrystallized.

**4.2. Comparison between field and lab**

**Comment:** This discussion should be linked to the comparison with Karulina et al. (2019). One important aspect I suggest you think about is cooling and then heating of the sea ice. We have tested relatively warm ice in-situ, then sampled cooled down (-15C) and stored (some weeks or some months), and finally heated again and tested. The samples that were cooled down and heated again were clearly stronger than the in-situ ice even if the temperature was the same! I think this is an important, and not understand mechanism in ice mechanics that should be studied, it may explain why SYI and Old Ice are both stronger than FYI even for comparable temperatures, and porosities.

**Response:** We will discuss more in the Section 4.2 based on your comment.

- (1) The sea ice flexural strength of our field measurements was 718.6±47.6 kPa. Karulina et al. (2019) reported a range of sea ice flexural strength of 109–415 kPa by performing full-scale tests in the Arctic regions. As stated before, the differences in sample preparation, size effects, and test techniques could produce different strength values between their and our tests.
- (2) As the reviewer commented, the thermal cycling of an ice sample would influence its mechanical behavior (Høyland, 2007). The sea ice block with warm in situ temperature was extracted and then cooled down for storage followed by heating again for final tests conducted in laboratory. While the samples that were cooled down and heated again are stronger than the in-situ ice even for comparable porosities. It is not clear why this is the case, but it is a possible reason that the strength estimated using equation derived from laboratory tests exceeds on site measured strength.

**Reference**

- Arakawa, M. and Maeno, N.: Mechanical strength of polycrystalline ice under uniaxial compression, Cold Reg. Sci. Technol., 26, 215–226, https://doi.org/ 10.1016/S0165-232X(97)00018-9, 1997.
- Bonath, V., Edeskär, T., Lintzén, N., Fransson, L., and Cwirzen, A.: Properties of ice from first-year ridges in the Barents Sea and Fram Strait, Cold Reg. Sci. Technol.,

168, 102890, https://doi.org/10.1016/j.coldregions.2019.102890, 2019.

- Cox, G. F. N. and Weeks, W. F.: Equations for determining the gas and brine volumes in sea ice samples, J. Glaciol., 29, 306–316, https://doi.org/:10.1017/S0022143000008364, 1983.
- Gold, L.: Statistical characteristics for the type and length of deformation-induced cracks in columnar-grain ice, J. Glaciol., 43, 311–320, https://doi.org/10.3189/S0022143000003269, 1997.
- Høyland, K. V.: Morphology and small-scale strength of ridges in the North-western Barents Sea, Cold Reg. Sci. Technol., 48, 169–187, https://doi.org/ 10.1016/j.coldregions.2007.01.006, 2007.
- Karulina, M., Marchenko, A., Karulin, E., Sodhi, D., Sakharov, A., and Chistyakov, P.: Full-scale flexural strength of sea ice and freshwater ice in Spitsbergen Fjords and North-West Barents Sea, Appl. Ocean Res., 90, 101853, https://doi.org/10.1016/j.apor.2019.101853, 2019.
- Kermani, M., Farzaneh, M., and Gagnon, R.: Bending strength and effective modulus of atmospheric ice, Cold Reg. Sci. Technol., 53, 162–169, https://doi.org/10.1016/j.coldregions.2007.08.006, 2008.
- Kuehn, G. and Schulson, E.: The mechanical properties of saline ice under uniaxial compression, Ann. Glaciol., 19, 39–48, https://doi.org/10.3189/1994AoG19-1-39-48, 1994.
- Moslet, P. O.: Field testing of uniaxial compression strength of columnar sea ice, Cold Reg. Sci. Technol., 48, 1–14, https://doi.org/10.1016/j.coldregions.2006.08.025, 2007.
- Schwarz, J., Frederking, R., Gavrillo, V., Petrov, I. G., Hirayama, K. I., Mellor, M., Tryde, P., and Vaudery, K. D.: Standardized testing methods for measuring mechanical properties of ice, Cold Reg. Sci. Technol., 4, 245–253, https://doi.org/10.1016/0165-232X(81)90007-0, 1981.
- Schulson, E. M.: Brittle failure of ice, Eng. Fract. Mech., 68, 1839–1887, https://doi.org/10.1016/S0013-7944(01)00037-6, 2001.
- Strub-Klein, L. and Høyland, K. V.: Spatial and temporal distributions of level ice properties: Experiments and thermo-mechanical analysis, Cold Reg. Sci. Technol., 71, 11–22, https://doi.org/ 10.1016/j.coldregions.2011.10.001, 2012.
- Sinha, N. K.: Crack-enhanced creep in polycrystalline material: strain-rate sensitive strength and deformation of ice, J. Mater. Sci., 23, 4415–4428, https://doi.org/10.1007/BF00551940, 1988.

- Timco, G. W. and Frederking, R. M. W.: A procedure to account for machine stiffness in uniaxial compression tests, in: Proceedings of the 7th IAHR International Symposium on Ice, Germany, 27–31 August 1984, 39–47, 1984.
- Timco, G. W. and Frederking, R. M. W.: Compressive strength of sea ice sheets, Cold Reg. Sci. Technol., 17, 227–240, https://doi.org/10.1016/S0165-232X(05)80003-5, 1990.
- Timco, G. W. and Weeks, W. F.: A review of the engineering properties of sea ice,ColdReg.Sci.Technol.,60,107–129,https://doi.org/10.1016/j.coldregions.2009.10.003, 2010.

---

## Author Comment (AC3)

**Response to the Comments of Reviewer3**

For the landfast sea ice in the Prydz Bay of East Antarctic, the flexural strength and uniaxial compressive strength were measured in field and in cold lab considering the influence of ice temperature, ice crystal size, loading rate and loading direction. Moreover, the brittle-ductile transition of sea ice in the uniaxial compression tests were discussed based on the experimental data. The measured results were analyzed comprehensively and compared with the literatures well. Some valuable data were obtained and can be applied in the engineering.

We would like to thank the reviewer for the helpful comments, based on which the manuscript has been revised accordingly. We have addressed the comments each below.

Some comments and suggestions are listed below for considerations.

**Comment:** Lines 153-154, How were the error propagations determined for the flexural strength, effective (elasticity) modulus, compressive strength and strain rate based on Eqs.(5) and (6)?

**Response:**

We will delete the *Section Uncertainty analysis* in the revised version. In the original manuscript, because the flexural strength, effective elastic modulus, compressive strength and strain rate are indirect measured variables, we had intended to estimate the uncertainties of these variables through error propagation equations. However, we found that we had missed the precondition that the direct measured variables used to derive indirect variables must be independent (e.g. the force and sample size are correlated). So, we will delete the section about uncertainty analysis.

**Comment:** "the effective modulus" should be "the effective Young's modulus" or "the effective modulus of elasticity".

**Response:** Thanks. We will correct it all in the text and figures.

**Comment:** Lines 180-184, the minimum flexural strength of mixed ice (511.3kPa) is higher than that of columnar ice (305.3kPa). This is quite different to the maximum and mean values. What is the main reason for the measured results?

**Response:** It is because of the differences of sea ice porosity. The sea ice porosity is

76.1–120.6‰ with an average of 90.6‰ for mixed ice, and is 43.3–168.6‰ with an average of 88.6‰ for columnar ice. Therefore, the minimum flexural strength of mixed ice (511.3 kPa) is higher than that of columnar ice (305.3 kPa); the maximum flexural strength of mixed ice (845.9 kPa) is lower than that of columnar ice (1119.7 kPa); the mean flexural strength of both types of samples is similar (687.9 vs. 698.8 kPa). We will also add the above explanation in the revised manuscript.

**Comment:** In Eqs.(9) and (10), please listed the dimensions for ice thickness h, the effective beam length r and the radius of loaded area c. Please check the other equations.
**Response:**
(1) The ice thickness was taken as 1.3 m, which is the thickness of congelation-ice layer of the ice block.
(2) Equations (9) and (10) work reliably when the loaded radius is not large enough compared with the characteristic length ($L_c$) of sea ice ($L_c = \left[ \frac{EH^3}{12k(1-v^2)} \right]^{\frac{1}{4}}$), and with sea ice porosity increasing from 40 to 260‰, the characteristic length decreased from 16.0 to 11.6 m; therefore, the loaded radii were selected as 2–10 m.
(3) As the effective beam length is an intermediate parameter, its dimensions were not shown in the text but will be given in the Fig. 14 in the revised version.
(4) Equations (9) and (10) just give the general method to calculate the extreme fiber stress in a cracked ice sheet under a uniformly distributed load. While it is Fig. 14 that shows the load that can be supported by landfast sea ice varying with different load radii. So, the dimensions of these parameters are given near Fig. 14.

---

## Author Response (AR1)

**Response to the Comments of Reviewer1**

This paper presents mechanical property test results of Antarctic sea ice and links those to the prevailing physical properties including porosity, brine volume, grain size, platelet spacing and strain rate. The paper contributes to the state of the art by providing valuable insights of the applicability of several existing methods to the estimation of Antarctic sea ice properties, specifically in the Prydz Bay, and by offering location-specific ice mechanical property and bearing capacity estimation for engineering purposes. The extensive effort to accomplish the research purpose is appreciated and the results are presented and analysed in a logical and clear manner.

We appreciate warmly for the reviewer's earnest work. The comments are constructive, and we have revised the manuscript accordingly. Detailed answers to all comments are provided below.

The specific comments are:

**Comment:** The brine volume and porosity were calculated using ice temperature, salinity and density using Cox and Weeks formulae. The calculation will most likely involve uncertainties which may have an impact on the later investigations. The authors are suggested to comment on the significance of this uncertainty source and its influence on the results of this work.

**Response:** Since it is not easy to quantify the uncertainties (the error propagation estimation needs independent direct measured variables, see response below), we stress this issue in a qualitative way as below.

It is noteworthy here that the calculations of brine volume fraction and porosity most likely involves uncertainties introduced by the measurement errors of ice physical properties; especially for sea ice porosity, the air volume fraction is largely dependent on ice density (Timco and Frederking, 1996).

The above statement also can be seen L133–136 in the revised manuscript.

**Comment:** Line 105: the authors are suggested to specify the speed of loading. It is not very clear what 'time-of-loading' means. I assume the ice beam fails very soon after loaded.

**Response:** We use strain rate to define loading speed in the new version. The strain rate

in three-point supported bending test is calculated using equation below (see Han et al. (2016))

$$\dot{\varepsilon}_{\mathrm{f}} = \frac{6h\dot{\delta}}{l^2}$$

where $\dot{\varepsilon}_{\mathrm{f}}$ is strain rate; $\dot{\delta}$ is deformation rate at beam midspan; $l$ is span between supports; $h$ is height of the beam. Result shows that the strain rate of our bending tests varies from $10^{-5}$ to $10^{-3}$ s$^{-1}$.

Please see L117 in the revised manuscript.

**Comment:** Line 199-200: some example references can be added to explain 'other commonly used functions'

**Response:** The relationship between sea ice flexural strength and square root of brine volume fraction has been generally reported in the exponential form (Timco and O'Brien, 1994; Karulina et al., 2019). Besides, the linear equation was also used in Krupina and Kubyshkin (2007). In this paper, we adopted more expressions including exponential, linear, logarithmic and power functions.

We have given the names of mathematical functions we used in the revised manuscript (L215) as we think it may be clear than giving the references.

**Comment:** The confidence intervals adopted for various analyse vary from 90% (e.g. Figure 7) to 99% (Figure 6). Is there a ration behind the selection of confidence intervals?

**Response:** The confidence intervals are determined according to the individual significance levels ($p$) obtained by regression analyses. For example, in Fig. 6, $p$ of the best-fit relationship between flexural strength and square root of porosity was less than 0.01, so we chose 99% as the confidence interval. In Fig. 7, $p > 0.1$ for the flexural strength-grain size best-fit equation, so the confidence interval was selected as 90%; and $p < 0.05$ for the flexural strength-platelet spacing best-fit equation, so the confidence interval was selected as 95%. Moreover, for the best-fit equations with various significant levels in different regimes, such as in Fig. 10a, we chose the maximum value as the final confidence interval for all the best-fit lines.

We have given a brief explanation in the figure titles, please see L221–222 and L319–320.

**Comment:** It would be helpful to indicate the range of salinity measured among the samples. It is found that the flexural strength is not sensitive to brine volume. Would it

be possible that this is because of the small range of salinity coverd by the samples (since they are from the same ice block)?

**Response:** Yes, what the reviewer suggested might be a reason. The salinity of congelation-ice samples in the bending tests was 1.0–5.1 psu. The brine volume fraction is a function of ice temperature, salinity, and density. The square root of brine volume fraction of our samples was 0.11–0.27, which is narrower than the ranges reported in Karulina et al. (2019) (0.16–0.39) and Timco and O'Brien (1994) (0–0.5), making the flexural strength of our samples not sensitive to brine volume fraction.

We have added the above discussion in the revised manuscript, please see L395–397.

**Comment:** How does Eq. (7) compare to the existing equations in the literature? Are they similar or do they differ a lot?

**Response:** To better compare our best-fit equation (Eq. 7) with existing equations reported in Karulina et al. (2019) and Timco and O'Brien (1994), the results of flexural strength calculated based on these equations were plotted against the square root of brine volume fraction in figure below. Results showed that the strength estimated using Karulina et al. (2019) was much lower than that estimated using ours. The strength estimated using Timco and O'Brien (1994) agreed better with ours, and only overestimated by 1.1 times.

We have shown the figure below in the revised manuscript as a subplot in Fig. 6 and given the above statement. Please see L226–229.

[Figure]

**Figure** Comparisons between estimated flexural strength using the empirical equations in previous studies and this work

**Comment:** Line 258-259: the sample size may be too small to draw the conclusion on temperature effect.

**Response:** The statement about the effect of ice temperature on snow-ice effective modulus has been deleted.

**Comment:** The first paragraph of 3.4.1: nice and thorough explanations are provided here to explain the measured trend of compressive strengths. More references are suggested here to support the reasoning, so that it does not look like own speculation. Same for later parts with such explanations.

**Response:** Thanks. More references have been cited to support the relative statements in the revised manuscript including Arakawa and Maeno (1997), Bonath et al. (2019), Gold (1997), Høyland (2007), Ji et al. (2020), Kuehn and Schulson (1994), Schulson (2001), Sinha (1988), and Strub-Klein and Høyland (2012).

**Comment:** The small size ice samples are cut from different positions along the thickness direction. Does the measured mechanical properties exhibit dependence on the thickness position? Typically congelation columnar ice is stronger at the top than at the bottom. This relates to Figure 14, where all the measurement has been plotted together in the same figure. The lower evelope probably corresponds mainly to flexural strength at the bottom, while in the case of bearing capacity ice fails at the top layer. This leads to conservative estimation of the bearing capacity.

**Response:**

(1) Due to the limited number of samples under each ice temperature and the focus of examining the effects of porosity and brine volume on sea ice strength, we did not record the thickness position of our bending samples in the whole ice sheet. Therefore, the dependence of strength on the ice depth is not able to be checked here. In general, as the reviewer said, the ice is stronger at the top than at the bottom.

(2) For estimating the bearing capacity of landfast sea ice, as the reviewer said, we conducted a conservative estimation. Because the real scenario is that the cargos are unloaded on the ice sheet, and thus, the strength of ice sheet is needed rather than that of small-scale samples. While the elastic modulus of sea ice varied along ice thickness, making it difficult to obtain the real distribution of stress along ice thickness. So, we conducted a conservative estimation for safe designing in this paper by adopting the minimum flexural strength. All the measured strength of ice samples was plotted in Fig. 15a, and the lower envelope of flexural strength was selected to represent the strength of ice sheet. The results indicate a minimum load that can put on ice.

(3) In addition, we think that the above estimation is close to the actual scenario to some degree. As the load is applied on ice sheet, the sheet should be compressed at the top and tensioned at the bottom. Ice is a material which is strong in compression

and weak in tension. So, the ice sheet deflects until the first crack or yielding develops in the underside of the sheet beneath the center of the load (Masterson, 2009). The low flexural strength often occurs at the bottom of ice sheet because of high ice temperature near freezing point; therefore, it is reasonable to use the lower envelope of flexural strength. The above discussion has been added in the revised manuscript, please see L483–487.

**Comment:** It may be worth also mentioning the influence of platelet spacing in the conclusion part.

**Response:** The statement below has been added in *Section Conclusions*:

The effects of sea ice sub-structure on columnar ice strength were investigated. Both flexural strength and effective elastic modulus increased with increasing platelet spacing, while the influence of grain size was not significant.

Please also see L515–517 in the new version.

Some technical corrections:

**Comment:** Line 51: the statement after 'because' tells why there are more understanding of mechanical properties of Arctic sea ice, but not really the reason why there are very few for the Antarctic. Consider rephrasing to make it more natural.

**Response:** The statement has been rephrased as below:

The mechanical properties of Arctic sea ice have been widely investigated in the last century because of booming oil and gas exploration in the Northern Hemisphere polar regions. While understanding of mechanical properties of Antarctic sea ice is limited due to less human and industry activities than those developed in the Arctic.

Please also see L53–55 in the revised manuscript.

**Comment:** Line 53: 'south pole' means exactly the pole (latitude 90). Here it should be something like 'Antarctic continent'.

**Response:** It has been replaced with Antarctic regions (L56).

**Comment:** Line 128: rule -> ruler?

**Response:** Yes, it is ruler. Corrected accordingly (L144).

**Comment:** Figure 4b: the pictures are small, making it difficult to see clearly the crystal

structures. Consider enlarging.

Response: A much clearer figure has been exhibited in the revised manuscript.

[Figure]

Comment: Eq. (8): typically equation follows immediately where it is firstly mentioned -> move 'overestimation ...' to after Eq. (8)
Response: Corrected accordingly (L327–331).

Comment: Line 379: empirical -> empirically
Response: Corrected accordingly (L437).

Comment: Line 420: photted -> photoed
Response: It has been changed to photo credit, please see L470.

Comment: Line 438: radiuses -> radii
Response: Corrected accordingly (L492).

References

Arakawa, M. and Maeno, N.: Mechanical strength of polycrystalline ice under uniaxial compression, Cold Reg. Sci. Technol., 26, 215–226, https://doi.org/10.1016/S0165-232X(97)00018-9, 1997.

Bonath, V., Edeskär, T., Lintzén, N., Fransson, L., and Cwirzen, A.: Properties of ice from first-year ridges in the Barents Sea and Fram Strait, Cold Reg. Sci. Technol., 168, 102890, https://doi.org/10.1016/j.coldregions.2019.102890, 2019.

Gold, L.: Statistical characteristics for the type and length of deformation-induced

cracks in columnar-grain ice, J. Glaciol., 43, 311–320, https://doi.org/10.3189/S0022143000003269, 1997.

Han, H., Jia, Q., Huang, W., and Li, Z.: Flexural strength and effective modulus of large columnar-grained freshwater ice, J. Cold Reg. Eng., 30, 04015005, https://doi.org/10.1061/(ASCE)CR.1943-5495.0000098, 2016.

Høyland, K. V.: Morphology and small-scale strength of ridges in the North-western Barents Sea, Cold Reg. Sci. Technol., 48, 169–187, https://doi.org/10.1016/j.coldregions.2007.01.006, 2007.

Ji, S., Chen, X., and Wang, A.: Influence of the loading direction on the uniaxial compressive strength of sea ice based on field measurements, Ann. Glaciol., 61, 86–96, https://doi.org/10.1017/aog.2020.14, 2020.

Karulina, M., Marchenko, A., Karulin, E., Sodhi, D., Sakharov, A., and Chistyakov, P.: Full-scale flexural strength of sea ice and freshwater ice in Spitsbergen Fjords and North-West Barents Sea, Appl. Ocean Res., 90, 101853, https://doi.org/10.1016/j.apor.2019.101853, 2019.

Krupina, N. A. and Kubyshkin, N. V.: Flexural strength of drifting level first-year ice in the Barents Sea, in: Proceedings of the 17th International Offshore and Polar Engineering Conference, Portugal, 1–6 July 2007, 2007

Kuehn, G. and Schulson, E.: The mechanical properties of saline ice under uniaxial compression, Ann. Glaciol., 19, 39–48, https://doi.org/10.3189/1994AoG19-1-39-48, 1994.

Masterson, D. M.: State of the art of ice bearing capacity and ice construction, Cold Reg. Sci. Technol., 58, 99–112, https://doi.org/10.1016/j.coldregions.2009.04.002, 2009.

Sinha, N. K.: Crack-enhanced creep in polycrystalline material: strain-rate sensitive strength and deformation of ice, J. Mater. Sci., 23, 4415–4428, https://doi.org/10.1007/BF00551940, 1988.

Schulson, E. M.: Brittle failure of ice, Eng. Fract. Mech., 68, 1839–1887, https://doi.org/10.1016/S0013-7944(01)00037-6, 2001.

Strub-Klein, L. and Høyland, K. V.: Spatial and temporal distributions of level ice properties: Experiments and thermo-mechanical analysis, Cold Reg. Sci. Technol., 71, 11–22, https://doi.org/ 10.1016/j.coldregions.2011.10.001, 2012.

Timco, G. W. and Frederking, R. M. W.: A review of sea ice density, Cold Reg. Sci. Technol., 24, 1–6, https://doi.org/10.1016/0165-232X(95)00007-X, 1996.

Timco, G. W. and O'Brien, S.: Flexural strength equation for sea ice, Cold Reg. Sci.

Technol., 22, 285–298, https://doi.org/10.1016/0165-232X(94)90006-X, 1994.

**Response to the Comments of Reviewer2**

An interesting paper, congratulations with all the good field work.

**Response:** We thank the reviewer for the recognition of our work. The comments are detailed and constructive, based on which we have revised the manuscript carefully. Please find our responses to individual comments below.

General comments

**Comment:** I suggest you try to keep the result to *your own results only*. The comparison to others and discussion on why fit better in the Discussion section. For example sub-section 3.2.1 is almost only comparison with others and discussions on why. Put this content in the Discussion section.

**Response:** Thanks. A new section *4.1 Comparisons with previous studies* has been added in the Discussion section, and the comparisons to other studies on flexural strength, effective modulus, and compressive strength have been moved to this section. Please see L376–419.

1. Introduction

**Comment:** OK, perhaps also refer to Strub-Klein and Høyland (2012).

**Response:** Their work has been cited. Please see L50–52.

2. In-situ sampling and laboratory experiments

2.1. In situ sampling

**Comment:** Ice temperature profile during field work? I suggest you move this information from section 4.2 into the *In-situ sampling* section.

**Response:** Corrected accordingly, please see L80–82.

**Comment:** What was the air temperature during field work? Do you have a air temperature history a few weeks back?

**Response:**

(1) During the sample preparation and tests, the air temperature varied from –2.6 to 1.8ºC with an average of –0.8±0.9ºC. We have added the information in the new version, please see L85–86.

(2) There is a weather station at the Zhongshan station. Since the field work site is not

far away from the Zhongshan station, so the air temperature recorded by the weather station is used. The figure below shows the air temperature from two months before field work. A rise in the air temperature occurred after 15 October 2019 (UTC) from below –10ºC to above –10ºC. This information and figure have also been added in the manuscript, please see L87–89 and Fig. 1.

[Figure]

Figure The air temperature from 1 October to 24 November 2019 at the Zhongshan station

**Comment:** How long time did the field work take? Or how long was the ice exposed to the air temperatures and possibly solar radiation?

**Response:** Approximately 2 hours after lifting onto the deck, part of the ice block was cut and machined into samples, and the bending tests were completed. During the tests, the air temperature varied from –2.6 to 1.8ºC with an average of –0.8±0.9ºC, and it was overcast with low solar radiation. We have given the above information in the revised manuscript, please see L84–87.

2.4. Bending tests

**Comment:** Elastic modulus. Could you explain how you derived these? Equation 2 only give a force and a displacement. But, there must be some kind of $\Delta F/\Delta \delta$? There are several ways to do this, one may search for the steepest part of the curve, use some kind of average etc.

**Response:** If the load is applied on the midspan of a simply supported beam, according to simple elastic beam theory, the midspan deflection of beam is

$$\delta = \frac{Fl^3}{4bh^3E}$$

where $\delta$ is midspan deflection, $F$ is force at failure, $E$ is Elastic modulus, $l$ is span between supports, $b$ and $h$ are section width and height of the beam.

In the three-point supported beam tests, with an assumption that the beam is perfectly elastic, the Elastic modulus can be then derived using Eq. (2) in our paper if $\delta$ is known. The equation was recommended by IAHR Section on Ice Problems (Schwarz et al., 1981) to provide guidelines for ice test methods and has been adopted by other reports (Karulina et al., 2019; Kermani et al., 2008). As sea ice is not turly elastic, and the derived modulus is termed effective modulus (Timco and Frederking, 2010). We have given a much clearer explanation on the equation, please see L121–125.

Additionally, according to the third reviewer's comment, the term of $E$ has been changed to effective elastic modulus.

2.5. Compression tests

**Comment:** Measure of displacement. I assume this is the position of the loaded plate and not the compression of the ice sample? I don't know your machine, but usually there is some elasticity in the machine that gives a somewhat lower compression of the sample than what is given by the displacement of the loaded plate.

**Response:** Yes, the measured displacement is the position of the loading plate. The deformation of the test machine results in a lower compression of the ice sample than the displacement of the loaded plate. Therefore, the nominal strain rate of the test is higher than the true strain rate of ice sample (Timco and Frederking, 1984), and the former was used in this paper. We used a universal testing machine to measure sea ice uniaxial compressive strength. The accurate stiffness of the machine was not measured. The machine was equipped with a portal frame with four columns supporting the upper beams, all of which are made of welded steel plates. So, it is expected to be rigid enough and produce a minor effect on the compressive tests.

More detailed information on our test machine (L146–148) and the effect of stiffness on strain rate (L159–165) have been added in the revised manuscript.

**Comment:** Equation 3. Perhaps use *Fmax*?

**Response:** Corrected accordingly, see L158–159.

2.6. Uncertainty analysis

**Comment:** The numbers here could be used to give a reasonable amount of numbers in the dervied properties.

**Response:** This section has been deleted in the revised manuscript because we found that we had missed the precondition of the error propagation that the direct measured

variables used to derive indirect variables must be independent. While the force applied and sample dimensions are correlated.

3. Results

3.1. Crystal structure

**Comment:** Where is the water line in Figure 4?

**Response:** The accurate ice freeboard was not measured in field as our focus was paid on snow and ice thickness. The freeboard was derived from a video recording the ice side surface by comparing with the snow thickness, and it was 19 cm approximately. We have given the freeboard in L171 and L183 and marked it in Fig. 4 in the revised manuscript.

3.2. Flexural strength

**Comment:** The values of flexural strength are given with a lot of number. But, if you consider an uncertainty of 0.002 and a value of about 700 it should be sufficient to give numbers like 511 kPA, 846 kPa etc.

**Response:** As the response to previous comment, the uncertainty analysis section has been deleted due to wrong understanding. While we agree with the reviewer that the strength value with integral number is enough considering the accuracy of force sensor and caliper. Therefore, we have rounded the values of flexural strength to integral number in the revised manuscript.

**Comment:** Why not give flexural strength of snow ice also here?

**Response:** The flexural strength of snow ice ranged from 93 to 177 kPa with an average of 123±37 kPa. The information has been added, please see L202–203.

**Comment:** As explained above I suggest to move the content of sub-section 3.2.1 (Congelation ice) to Discussion.

**Response:** Corrected accordingly, see *Section 4.1.1*.

**Comment:** Line 192. *the region specific*. I don't like this explanation at all. The ice does not know where it is, it only knows which physical conditions it has been exposed to. It is OK if you cannot explain why things are different, but do not blindly blame Geography!

**Response:** Thanks for your comment. We have deleted the original explanation and

rephrased it according to your comment below. Please see L382–386.

**Comment:** Differences to Timco and O'Brien (1994). T&B give some kind of upper limit and this means that almost any set of experiments will give lower average values. In other words it is natural that you find lower values.

**Response:** Discussion has been added following your suggestion in the revised manuscript, please see L386–388.

**Comment:** Differences to Karulina et al. (2019). Here your results are higher and there are some obvious differences that should be discussed. Firstly, Karulina et al. (2019) tested in field, secondly they tested larger beams larger beams. It could be that their beams had more weaknesses than yours. You prepared the beams carefully in the lab and these two facts may help to explain. Also the different testing methods may have contributed.

**Response:** We have revised the discussion based on your comments as below.

The differences may be attributed to several facts. The first is that our tests were conducted in the laboratory, where samples were prepared with caution, while the tests in Karulina et al. (2019) were performed in field. Further, the ice samples in Karulina et al. (2019) were much larger and contained more potential weaknesses than ours. Besides, the flexural strength in Karulina et al. (2019) was derived using cantilever beams, and stress concentrations at the root of beam resulted in low strength.

The above statement can also be seen in L382–386 in the revised manuscript.

**Comment:** It is interesting and new that you investigate the flexural strength in relation to grain size and platelet spacing. Very nice.

**Response:** Thanks for the reviewer's approval on this work.

**Comment:** Figure 5. It is interesting to note that the slope was more or less equal for the columnar and mixed ice, in spite of different strengths. And that the peak deformation was equal for the snow ice and the mixed ice in spite of very different strengths. Was this coincidental?

**Response:** We have checked all our data and found that:

(1) The mean slopes of force varying with deformation were similar between columnar-ice samples ($512\pm246$ N·mm$^{-1}$) and mixed-ice samples ($625\pm178$ N·mm$^{-1}$).

(2) The mean peak deflections of samples at failure were similar between snow-ice

samples (0.31±0.11 mm) and mixed-ice samples (0.41±0.21 mm).

We have added the above information in the revised manuscript, please see L192–194.

**3.3. Effective modulus**

**Comment:** As explained above you need to explain how you found the effective elastic modulus ($E$).

**Response:** Thanks, please see the above response where we have made a detailed explanation.

**Comment:** $E$ is a function of force, displacement and time. The more time a tests takes the more important becomes the viscous (or delayed viscous) deformation. The time-dependent deformation is know to be a function of salinity(brine volume). Did Karulina et al. (2019) load with the same load/displacement rate as you did? If they loaded more slowly it may explain why they found $E = f$ (b$rinevolume$)?

**Response:**

(1) Based on the first reviewer's comment, we have calculated the strain rate of our bending tests, giving a range between $10^{-5}$ to $10^{-3}$ s$^{-1}$.

(2) The strain rate in Karulina et al. (2019) varied from $10^{-4}$ to $10^{-3}$ s$^{-1}$. So, our tests were performed at a similar or even slower rate than Karulina et al. (2019), indicating that strain rate seems may not be the factor judging whether effective elastic modulus is dependent on brine volume or porosity. We have added the above discussion in the revised manuscript, please see L398–403.

**3.4. Uniaxial compressive strength**

**3.4.1. Congelation ice**

**Comment:** There is much more available published data on uni-axial strength and it is good to see that your results are more or less in line with what we think we know from before.

**Response:** In the revised manuscript, we have cited more references and made brief statements on these similarities.

(1) For stress-strain curves, Bonath et al. (2019) and Arakawa and Maeno (1997) have been commented and cited (L286–287).

(2) For the failure modes compressed under different directions, Gold (1997), Sinha (1988) and Kuehn and Schulson (1994) have been cited (L293 and 295).

(3) For the effect of stain rate on compressive strength and corresponding fitting

equations, Schulson (2001), Timco and Frederking (1990), Arakawa and Maeno (1997), and Høyland (2007) have been cited (L306, L307, L309, L311, and L312).

(4) For the comparisons between horizontally- and vertically loaded strength, Strub-Klein and Høyland (2012) has been cited (L315).

(5) For the effect of sea ice porosity on compressive strength, Moslet (2007) has been commented and cited (L323–324).

**3.4.2. Mixed and snow ice**

**Comment:** Any comment on physical properties of the snow ice? You do not report densities or porosities. Why? If the ice was too porous to shape samples properly, please say so. Did you have any impression from visual observation? Was the ice more porous or why was it weaker?

**Response:**

(1) The mean density of snow-ice samples in bending test was $0.55\pm0.01$ g·cm$^{-3}$, and that in compression test was $0.61\pm0.13$ g·cm$^{-3}$. The snow-ice samples could be distinguished easily by their white appearance and light weight.

(2) As snow ice was not formed by congelation of sea water, the equations proposed by Cox and Weeks (1983) were not able to be used to determine porosity. From visual observation, the snow ice was relatively porous than the underlying congelation ice. The snow ice was more compacted than the new snow on the surface; therefore, it can still be machined into regular shape.

We have added the above information in the revised manuscript, please see L77–78, L186–187, L276–277.

**4. Discussion**

**4.1. Ratios between strengths**

**Comment:** You could also compare with Moslet (2007) and Strub-Klein and Høyland (2012), they also report vertical / horizontal uni-axial compression strengths. I don't think you can claim that you have found the unique ratios between uni-axial compression in vertical direction, the same in horizontal direction and flexural strengths. Moslet (2007) argues that this is a function of ice temperature among other things.

**Response:** We have compared our ratios with those reported in Moslet (2007) and Strub-Klein and Høyland (2012) in the revised manuscript. It was found that the ratio of vertically to horizontally loaded uniaxial compressive strength in our tests was independent of porosity, and the average was $3.1\pm0.9$. The ratio of vertically loaded

uniaxial compressive strength to flexural strength decreased with increasing porosity, and it reached 8.0 for sea ice with small porosity and 4.0 for sea ice with large porosity approximately. The average ratio was 7.4±1.9. Moslet (2007) reported ratios between vertically and horizontally loaded strength of columnar ice of 1.3 for cold ice ($<-10$ºC) and of 4–5 for warm ice. Strub-Klein and Høyland (2012) reported low vertically-to-horizontally loaded strength ratios of 1.4–1.8 for granular and columnar ice, probably because the ice cover where test samples were removed had already been broken and recrystallized before sampling

The above discussion can also be seen in L425–428 and L432–436 in the revised manuscript.

4.2. Comparison between field and lab

**Comment:** This discussion should be linked to the comparison with Karulina et al. (2019). One important aspect I suggest you think about is cooling and then heating of the sea ice. We have tested relatively warm ice in-situ, then sampled cooled down (-15C) and stored (some weeks or some months), and finally heated again and tested. The samples that were cooled down and heated again were clearly stronger than the in-situ ice even if the temperature was the same! I think this is an important, and not understand mechanism in ice mechanics that should be studied, it may explain why SYI and Old Ice are both stronger than FYI even for comparable temperatures, and porosities.

**Response:** We have made more discussion in the Section 4.3 based on your comment.

(1) The sea ice flexural strength of our field measurements was 719±48 kPa. Karulina et al. (2019) reported a range of sea ice flexural strength of 109–415 kPa by performing full-scale tests in the Arctic regions, which is lower than our field measured strengths. Our field measurements were also performed using small-scale three-point supported beam tests, as stated before, the differences in size effects and test techniques could result in different strength values.

(2) As the reviewer commented, the thermal cycling of an ice sample would influence its mechanical behavior (Høyland, 2007). The sea ice with warm in situ temperature was sampled and then cooled down for storage followed by heating again for final tests completed in laboratory. While the samples that were cooled down and heated again are stronger than the in situ ice even for comparable porosities. Currently this phenomenon is not clear, and requires further studies in future.

The above discussion can be found in L444–453 in the revised manuscript.

References

Arakawa, M. and Maeno, N.: Mechanical strength of polycrystalline ice under uniaxial compression, Cold Reg. Sci. Technol., 26, 215–226, https://doi.org/10.1016/S0165-232X(97)00018-9, 1997.

Bonath, V., Edeskär, T., Lintzén, N., Fransson, L., and Cwirzen, A.: Properties of ice from first-year ridges in the Barents Sea and Fram Strait, Cold Reg. Sci. Technol., 168, 102890, https://doi.org/10.1016/j.coldregions.2019.102890, 2019.

Cox, G. F. N. and Weeks, W. F.: Equations for determining the gas and brine volumes in sea ice samples, J. Glaciol., 29, 306–316, https://doi.org/:10.1017/S0022143000008364, 1983.

Gold, L.: Statistical characteristics for the type and length of deformation-induced cracks in columnar-grain ice, J. Glaciol., 43, 311–320, https://doi.org/10.3189/S0022143000003269, 1997.

Høyland, K. V.: Morphology and small-scale strength of ridges in the North-western Barents Sea, Cold Reg. Sci. Technol., 48, 169–187, https://doi.org/10.1016/j.coldregions.2007.01.006, 2007.

Karulina, M., Marchenko, A., Karulin, E., Sodhi, D., Sakharov, A., and Chistyakov, P.: Full-scale flexural strength of sea ice and freshwater ice in Spitsbergen Fjords and North-West Barents Sea, Appl. Ocean Res., 90, 101853, https://doi.org/10.1016/j.apor.2019.101853, 2019.

Kermani, M., Farzaneh, M., and Gagnon, R.: Bending strength and effective modulus of atmospheric ice, Cold Reg. Sci. Technol., 53, 162–169, https://doi.org/10.1016/j.coldregions.2007.08.006, 2008.

Kuehn, G. and Schulson, E.: The mechanical properties of saline ice under uniaxial compression, Ann. Glaciol., 19, 39–48, https://doi.org/10.3189/1994AoG19-1-39-48, 1994.

Moslet, P. O.: Field testing of uniaxial compression strength of columnar sea ice, Cold Reg. Sci. Technol., 48, 1–14, https://doi.org/10.1016/j.coldregions.2006.08.025, 2007.

Schwarz, J., Frederking, R., Gavrillo, V., Petrov, I. G., Hirayama, K. I., Mellor, M., Tryde, P., and Vaudery, K. D.: Standardized testing methods for measuring mechanical properties of ice, Cold Reg. Sci. Technol., 4, 245–253, https://doi.org/10.1016/0165-232X(81)90007-0, 1981.

Schulson, E. M.: Brittle failure of ice, Eng. Fract. Mech., 68, 1839–1887, https://doi.org/10.1016/S0013-7944(01)00037-6, 2001.

Strub-Klein, L. and Høyland, K. V.: Spatial and temporal distributions of level ice properties: Experiments and thermo-mechanical analysis, Cold Reg. Sci. Technol., 71, 11–22, https://doi.org/ 10.1016/j.coldregions.2011.10.001, 2012.

Sinha, N. K.: Crack-enhanced creep in polycrystalline material: strain-rate sensitive strength and deformation of ice, J. Mater. Sci., 23, 4415–4428, https://doi.org/10.1007/BF00551940, 1988.

Timco, G. W. and Frederking, R. M. W.: A procedure to account for machine stiffness in uniaxial compression tests, in: Proceedings of the 7th IAHR International Symposium on Ice, Germany, 27–31 August 1984, 39–47, 1984.

Timco, G. W. and Frederking, R. M. W.: Compressive strength of sea ice sheets, Cold Reg. Sci. Technol., 17, 227–240, https://doi.org/10.1016/S0165-232X(05)80003-5, 1990.

Timco, G. W. and Weeks, W. F.: A review of the engineering properties of sea ice, Cold Reg. Sci. Technol., 60, 107–129, https://doi.org/10.1016/j.coldregions.2009.10.003, 2010.

For the landfast sea ice in the Prydz Bay of East Antarctic, the flexural strength and uniaxial compressive strength were measured in field and in cold lab considering the influence of ice temperature, ice crystal size, loading rate and loading direction. Moreover, the brittle-ductile transition of sea ice in the uniaxial compression tests were discussed based on the experimental data. The measured results were analyzed comprehensively and compared with the literatures well. Some valuable data were obtained and can be applied in the engineering.

We would like to thank the reviewer for the helpful comments, based on which the manuscript has been revised accordingly. We have addressed the comments each below.

Some comments and suggestions are listed below for considerations.

**Comment:** Lines 153-154, How were the error propagations determined for the flexural strength, effective (elasticity) modulus, compressive strength and strain rate based on Eqs.(5) and (6)?

**Response:** We have deleted the *Section Uncertainty analysis* in the revised manuscript. In the original manuscript, the flexural strength, effective elastic modulus, compressive strength and strain rate are indirect measured variables, so we had intended to estimate the uncertainties of these variables through error propagation equations. However, we found that we had missed the precondition that the direct measured variables used to derive indirect variables must be independent (e.g. the force and sample size are correlated). So, we have deleted the section about uncertainty analysis.

**Comment:** "the effective modulus" should be "the effective Young's modulus" or "the effective modulus of elasticity".

**Response:** Thanks. We have corrected it all in the text and figures.

**Comment:** Lines 180-184, the minimum flexural strength of mixed ice (511.3kPa) is higher than that of columnar ice (305.3kPa). This is quite different to the maximum and mean values. What is the main reason for the measured results?

**Response:** It is because of the inverse relationship between sea ice strength and porosity. The sea ice porosity was 76.1–120.6‰ with an average of 90.6‰ for mixed ice, and

was 43.3–168.6‰ with an average of 88.6‰ for columnar ice. Therefore, the minimum flexural strength of mixed ice (511.3 kPa) was higher than that of columnar ice (305.3 kPa); the maximum flexural strength of mixed ice (845.9 kPa) was lower than that of columnar ice (1119.7 kPa); the mean flexural strength of both types of samples was similar (687.9 vs. 698.8 kPa). We have added the above explanation in the revised manuscript, please see L199–203.

**Comment:** In Eqs.(9) and (10), please listed the dimensions for ice thickness h, the effective beam length r and the radius of loaded area c. Please check the other equations.
**Response:**

(1) The ice thickness was taken as 1.3 m, which is the thickness of congelation-ice layer of the ice block (see L489–490).

(2) Equations (9) and (10) worked reliably when the loaded radius was not large enough compared with the characteristic length ($L_c$) of sea ice ($L_c = \left[\dfrac{EH^3}{12k(1-v^2)}\right]^{\frac{1}{4}}$), and with sea ice porosity increasing from 40 to 260‰, the characteristic length decreased from 16.0 to 11.6 m Therefore, the loaded radii were selected as 2–10 m (see L492–493).

(3) As the effective beam length is an intermediate parameter in the calculation of ice bearing capacity, we have not shown it in the text but given it in the Fig. 15 in the revised version (see L499).

(4) Equations (9) and (10) just give the general method to calculate the extreme fiber stress in a cracked ice sheet under a uniformly distributed load. While it is Fig. 15 that shows the load that can be supported by landfast sea ice varying with different load radii. So, the dimensions of these parameters are given near Fig. 15.

---

## Author Response (AR2)

**Response to the Comments of Editor**

I am happy to accept your manuscript to be published in TC. Thank you for your contribution and congratulations.

Before final publication. Please consider adding one vertical bar line or a visible mark on Fig1d to indicate which day the ice sample was collected.

**Response:**

We appreciate all the earnest work made by editor Bin Cheng and thank for his time and consideration.

As for the technical correction proposed by the editor, we have given a visible mark with corresponding explanation in the figure (please see below). The information of sampling date (24 November 2019, UTC) has also been added in the text, please see L76–77.

[Figure]

Figure The revised Fig. 1